# Microglia-specific overexpression of α-synuclein leads to severe dopaminergic neurodegeneration by phagocytic exhaustion and oxidative toxicity

Simone Bido [1], Sharon Muggeo [1], Luca Massimino[1], Matteo Jacopo Marzi[2], Serena Gea Giannelli[1], Elena Melacini[1,3], Melania Nannoni[1], Diana Gambarè[1], Edoardo Bellini [1], Gabriele Ordazzo[1], Greta Rossi[1], Camilla Maffezzini[1], Angelo Iannelli[1,3], Mirko Luoni[1], Marco Bacigaluppi[4], Silvia Gregori[5], Francesco Nicassio [2] & Vania Broccoli [1,3✉]

Recent findings in human samples and animal models support the involvement of inflammation in the development of Parkinson's disease. Nevertheless, it is currently unknown whether microglial activation constitutes a primary event in neurodegeneration. We generated a new mouse model by lentiviral-mediated selective α-synuclein (αSYN) accumulation in microglial cells. Surprisingly, these mice developed progressive degeneration of dopaminergic (DA) neurons without endogenous αSYN aggregation. Transcriptomics and functional assessment revealed that αSYN-accumulating microglial cells developed a strong reactive state with phagocytic exhaustion and excessive production of oxidative and proinflammatory molecules. This inflammatory state created a molecular feed-forward vicious cycle between microglia and IFNγ-secreting immune cells infiltrating the brain parenchyma. Pharmacological inhibition of oxidative and nitrosative molecule production was sufficient to attenuate neurodegeneration. These results suggest that αSYN accumulation in microglia induces selective DA neuronal degeneration by promoting phagocytic exhaustion, an excessively toxic environment and the selective recruitment of peripheral immune cells.

[1] Division of Neuroscience, IRCCS San Raffaele Scientific Institute, 20132 Milan, Italy. [2] Center for Genomic Science of IIT@SEMM, Istituto Italiano di Tecnologia (IIT), 20139 Milan, Italy. [3] National Research Council (CNR), Institute of Neuroscience, 20129 Milan, Italy. [4] Division of Neuroscience, Institute of Experimental Neurology, IRCCS San Raffaele Scientific Institute, Milan, Italy. [5] San Raffaele Telethon Institute for Gene Therapy, IRCCS San Raffaele Scientific Institute, Milan, Italy. ✉email: broccoli.vania@hsr.it

Microglia are brain resident immune cells that continuously patrol the neural parenchyma for preventing or counteracting microenvironmental homeostasis alterations and represent the main source of immune modulators in the brain[1–3]. Noxious agents or injurious processes can trigger activation of microglial cells, inducing their proliferation, morphological changes and release of immune regulators, growth factors, and neurotoxic substrates. In the presence of enduring pathological stimuli, microglia can be stabilized in a chronic proinflammatory phenotype which can become detrimental exacerbating pathological processes[4,5]. Moreover aging, which represents the main risk factor for neurodegenerative diseases, is a condition strongly associated with sustained activation of microglial cells and enhanced release of pro-inflammatory mediators in the central nervous system and periphery[6,7]. Despite the pivotal role of microglial inflammation already described in Alzheimer's disease and multiple sclerosis, its exact contribution to the pathogenetic mechanisms underlying Parkinson's disease (PD) is debated[8–10]. PD is characterized by motor and cognitive impairments caused by the progressive loss of the dopaminergic neurons in the *substantia nigra pars compacta*[11]. Even though the clinical manifestation of PD is well described, the pathophysiological mechanisms underlying PD etiology remains not fully understood[12–14]. The reason for this uncertainty lies in the multifactorial nature of the disease which beyond genetic factors includes environmental triggers and aging. In human postmortem tissues and PD animal models, there is growing evidence for reactive microgliosis in the brain nigral tissue associated with increased expression of proinflammatory cytokines and infiltration of macrophages[15–17]. The heightened expression of major histocompatibility complex (MHC) class II molecules on macrophages/microglia in both human and murine PD brains has also been documented[18]. Moreover, Rayaprolu and colleagues reported that the p.R47H substitution on the microglia-expressed gene TREM2 represents a risk factor for frontotemporal dementia and PD[19]. Progressive aggregation of α-synuclein (αSYN) in large deposits, named as Lewy bodies (LBs), mainly localized within neurons across the brain, is a PD neuropathological hallmark of PD. The disease condition induces αSYN self-aggregation with the generation of insoluble fibrillar species that are the main components of LBs within the neurons[20,21]. However, misfolded αSYN can be released from neurons and propagate to neighboring neuronal and glial cells where it stimulates the endogenous protein aggregation leading to the propagation of the αSYN pathology[22,23]. αSYN spreading is associated with microglial activation and sustained production of pro-inflammatory mediators[24,25]. Hence, chronic inflamed microglia have an active role in facilitating the spreading of aggregating-prone αSYN, releasing toxic compounds and aberrant synapse pruning[26,27]. However, the exact impact of reactive microglia in exacerbating and amplifying the pathological processes remains to be defined. Intranigral lipopolysaccharide (LPS) injections trigger activation of microglia and subsequent progressive loss of DA neurons[28,29]. These findings reinforced the association between neuroinflammation and DA neuronal degeneration. However, LPS mimics generic pathogen-induced inflammation further stimulating astrocytes and myeloid cells and how its action recapitulates the pathological mechanisms exerted by αSYN accumulation is not fully unveiled yet[30].

More generally, the prevailing view is that neuroinflammation is a consequential phenomenon in PD which follows the primary cause of neurodegeneration and acts as a multifaceted modulator of the disease state progression. Based on this concept, interventions to reduce the general neuroinflammation state have in principle clinical relevance. Thus, the non-steroidal anti-inflammation drugs are being tested in selected clinical trials with PD

patients[31,32]. However, considering the general protective role played by microglial cells, these treatments are expected to raise some troublesome side effects since implicate a long-term repression of the microglial activity. Therefore, a more precise description of the consequences directly induced by the diseased microglia in reliable PD models and the exact contribution of their intrinsic molecular pathways is of paramount importance to direct new drug discovery efforts for developing new treatments for PD.

Thus, to better determine the primary role of microglial cells in PD onset and progression and to describe the effects triggered by the diseased microglia in an otherwise healthy tissue, we developed a mouse model with selective αSYN accumulation in nigral microglia. With this approach we unveiled the primary neurotoxicity of these cells which caused the selective degeneration of the surrounding DA neurons. This degenerative process is stimulated by a pathological self-reinforcing cycle between the exhaustion of phagocytosis and the significant stimulation of oxidative genes which leads to a strong release of oxidative and nitrosative toxic species creating a toxic local environment. We also provided a description at single-cell resolution of the infiltrated immune cells and the αSYN-accumulating microglia which identified immune cell populations whose pathological significance will be addressed in the future using this experimental system.

## Results

**Microglia-specific expression of human *SNCA*[G420A] by cre-inducible lentiviral-mediated expression**. To ascertain the possibility of expressing a transgene efficiently and selectively in microglial cells and DA neurons, we generated both Cre-inducible AAV and lentiviral vectors with the GFP coding region in a FLEX configuration[33] (Fig. 1 and Supplementary Fig. 1a). With this system, the fluorescence reporter is expected to be activated only in Cre-expressing cells by inverting the transgene in the sense direction for productive transcription. We generated AAV and lentiviral particles with inducible vectors and separately delivered them into the *substantia nigra* of CX3CR1-CreERT2 and DAT-Cre mice at the concentration of 10E10 vg and 10E8 PFU, respectively, to assess transgene expression at the final concentration. Two weeks later, GFP fluorescence was examined by colocalization with TH and IBA1 to label DA neurons and microglia, respectively. Interestingly, the AAV-induced GFP signal was detectable in many TH-negative cells in DAT-Cre mouse brain tissue (Supplementary Fig. 1a–j). Similarly, several brain cells other than microglia were GFP + in AAV-injected CX3CR1-CreERT2 mice, while microglia did not express the transgene, confirming microglial resistance to AAV infection as described previously[34,35]. Similar ectopic expression of the transgene was found using decreasing titers of AAV (10E9 and 10E8 vg; Supplementary Fig. 1a–j). In contrast, GFP expression driven by lentiviral transduction (10E8 PFU) was detected in DAT-Cre and CX3CR1-CreERT2 mice exclusively in TH + neurons and microglia, respectively, as confirmed by double and triple immunostaining (Supplementary Figs. 1k–q and 2a–h). These findings indicated that high specificity of the FLEX cassette conditional system was only achieved when the system was packaged in lentiviruses. Given these results, we opted for the lentiviral Cre system to drive conditional expression of the human *SNCA*[G420A] allele encoding the αSYN-A53T mutant form in vivo[36].

**SNCA[G420A]-overexpressing microglia are highly activated and accumulate pS129αSYN[+] aggregates**. Ef1a-FLEX-SNCA[G420A] lentiviruses were delivered to the *substantia nigra* by stereotactic

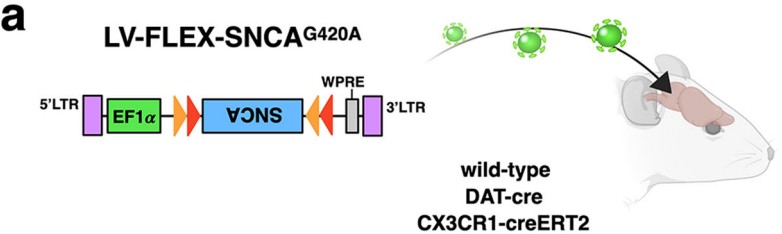

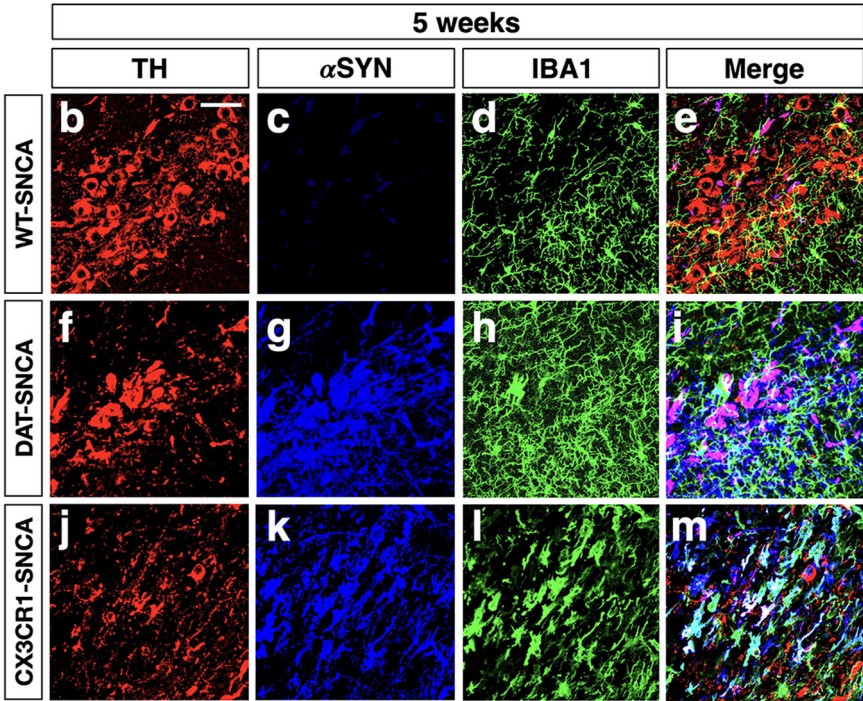

**Fig. 1 Cell-type selective overexpression of *SNCA*^G420A in vivo through lentiviral gene transfer. a** Illustration of the experimental design to chieve cell-type-specific αSYN accumulation. The lentiviral vector carrying the Cre-inducible transgene *SNCA*^G420A is inoculated through stereotaxic injections into the brain nigral tissue of wild-type, DAT-Cre and CX3CR1-CreERT2 mice for restricted transgene expression on TH and IBA1 positive cells, respectively. Control condition is represented by wild-type animals injected with LV-FLEX-SNCA^G420A (WT-SNCA, only for Fig. 1). The LB509 antibody recognizing only the human, but not the mouse, αSYN is used to selectively detect the viral exogenous protein. **b–m** Representative images of $n = 8$ independent nigrae showing the conditional viral *SNCA*^G420A gene expression and the localization of TH and IBA1 that identify DA neurons and myeloid cells, respectively. **b–e** In transduced wild-type animals the transgene expression is undetectable. **f–i** In DAT-SNCA animals αSYN expression (blue) is limited to the TH^+ neurons (red). **j–m** Conversely, transduced CX3CR1-SNCA mice show αSYN localization exclusively in IBA1^+ myeloid cells (red). Scale bar: 45 μm.

injections in 8-week-old CX3CR1-CreERT2 (CX3CR1-SNCA) and DAT-cre (DAT-SNCA) mice. As a control condition, we injected CX3CR1-CreERT2 mice with Cre-inducible lentiviruses that express destabilized GFP (CX3CR1-dGFP), hereinafter simply referred to as the control. We excluded the use of GFP since it is toxic at least to DA neurons (Supplementary Fig. 3a). In this context, the control mice were subjected to the same surgical manipulations, and the targeted brain populations expressed similar levels of the transgenes and their relative protein products (Supplementary Fig. 4a–c). Initial analysis was performed 5 weeks after viral administration to assess *SNCA*^G420A expression and αSYN accumulation and correlate these findings with microglial status and phenotype. First, human αSYN-specific immunostaining was performed and showed colocalization of αSYN with ~55% of the IBA1 signal in the nigra, indicating efficient lentiviral transduction that induced wide expression of *SNCA*^G420A in the microglial cell compartment (Supplementary Fig. 2i–l). However, it is plausible that additional microglial cells expressing low levels of αSYN were not captured by this particular staining. Next, we

examined whether αSYN-accumulating microglia exhibited signs of pathological αSYN aggregation. Double immunostaining revealed that pS129αSYN colocalized with nearly 40% of αSYN staining, indicating ongoing pathological αSYN protein aggregation (Fig. 2 and Supplementary Fig. 2m–p). Morphological and fractal analysis showed that infected microglia exhibited a strong retraction and increased process complexity with the acquisition of hypertrophic morphology and augmented cell size, suggesting significant polarization toward an activated state (Fig. 2). Accordingly, a marked fraction of αSYN^+ microglial cells were positive for CD68, a key marker of activated cells (Fig. 2j). In DAT-SNCA mice, ~55% of *SNCA*^G420A expressing DA neurons developed diffuse pS129αSYN positivity (Fig. 2b). In these mice, microglial cells developed a ramified morphology with an increased number of shorter processes, and only a minimal fraction of these cells expressed low levels of CD68 (Fig. 2e, i, m, p). Accordingly, reactive GFAP + astrocytes exhibiting hypertrophic cellular processes were evident only in CX3CR1-SNCA mice but not in DAT-SNCA mice (Supplementary Fig. 5). Collectively,

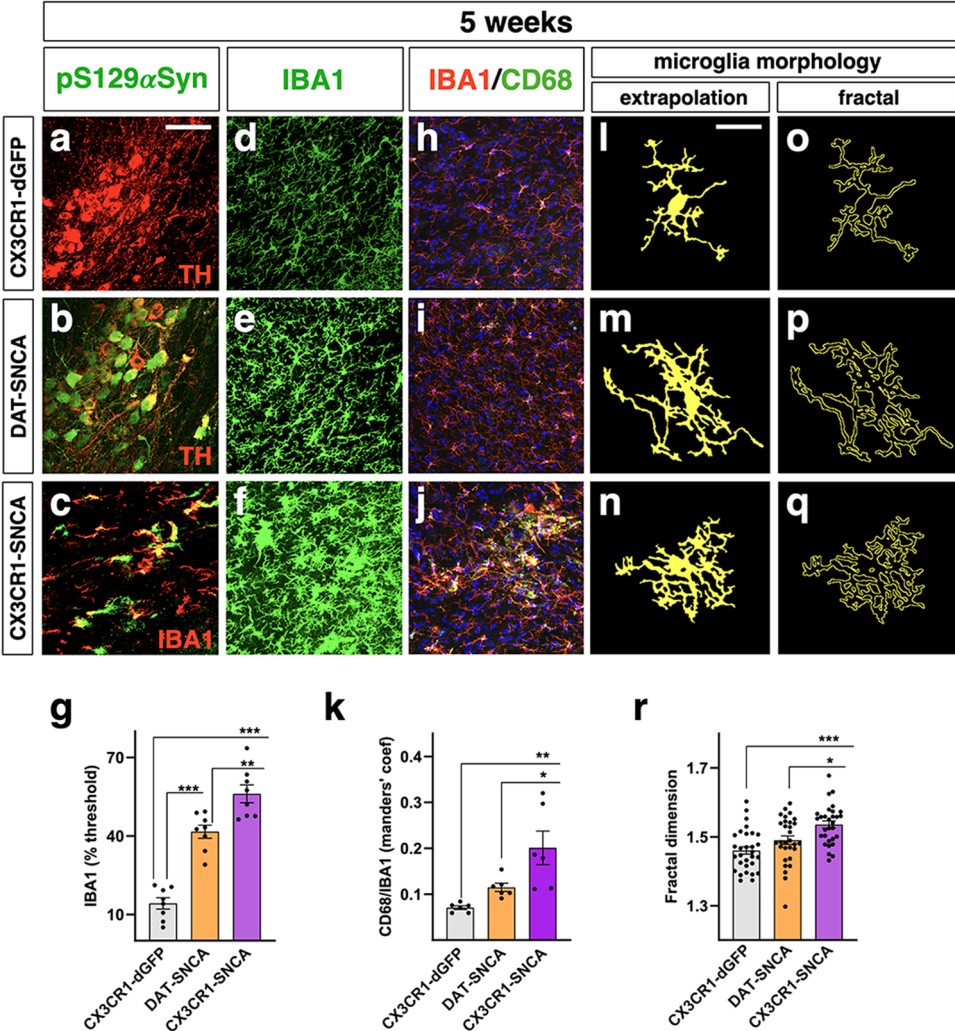

**Fig. 2 Microglia accumulating αSYN show a reactive state. a–c** αSYN expressing cells display diffuse accumulation of pS129αSYN (green), which is limited to IBA1 and TH positive cells in CX3CR1- and DAT-cre animals, respectively (representative images of $n = 8$ independent nigrae). **d–g** IBA1 signal, measured as the occupied area by a fixed signal threshold, is increased after αSYN upregulation, with CX3CR1-SNCA animals exhibiting the greater increase when compared to DAT-SNCA mice. Data are the mean ± s.e.m from $n = 8$ nigrae. One-way ANOVA is followed by Bonferroni's post-test. **p = 0.0034 and ***p < 0.0001. Scale bar, 45 μm. **h–k** Colocalization index (Manders' coefficient) between CD68 and IBA1 shows the increased colocalization in CX3CR1-SNCA mice with no significant augmentation in control and DAT-SNCA groups ($p = 0.52$). Data are the mean ± s.e.m from $n = 6$ nigrae. One-way ANOVA is followed by Bonferroni's post-test. *p = 0.0034 and **p < 0.0001. Scale bar, 45 μm. **l–r** The morphology measurement achieved by fractal analysis reveals the increased complexity of microglia shape in DAT-SNCA and CX3CR1-SNCA animals, with CX3CR1-SNCA showing the greater level of complexity. Data are the mean ± s.e.m from $n = 30$ cells from six different nigrae. One-way ANOVA is followed by Bonferroni's post-test. *p = 0.0156 and ***p < 0.0001. Scale bar, 15 μm.

5 weeks after viral transduction, microglial cells were strongly activated in CX3CR1-SNCA mice, but much less so in DAT-SNCA mice, as indicated by changes in morphology, behavior and key marker expression. Subsequent analysis showed that activated microglia were also detectable in CX3CR1-SNCA mice at 2 and 15 weeks after treatment, suggesting a fast and durable inflammatory response after microglia-specific $SNCA^{G420A}$ expression (Supplementary Fig. 6). In addition, microglia overexpressing the native GFP form after LV-FLEX-GFP injection in CX3CR1-creERT2 mice did not develop any sign of reactivity, corroborating the αSYN-specific detrimental effects in microglia (Supplementary Fig. 3b–k).

**Microglia-specific αSYN accumulation leads to selective nigral DA neurodegeneration.** Next, we wondered whether restricted αSYN accumulation in nigral microglia could induce any effect on DA neurons over time. Thus, the total number of nigral DA

neurons was evaluated in DAT-SNCA, CX3CR1-SNCA and control mice by staining for tyrosine hydroxylase (TH) and unbiased stereology at 2, 5, and 15 weeks after $SNCA^{G420A}$ transgene activation (Fig. 3). As anticipated, DAT-SNCA mice showed progressive DA neuronal degeneration, with ~28% and 92% loss of TH-immunopositive nigral cells relative to those of control mice at 5 and 15 weeks post-transduction, respectively (Fig. 3d, f, h, i). Unexpectedly, CX3CR1-SNCA mice showed incremental losses in both nigral and VTA DA neurons over time (Fig. 3d, f). In fact, we quantified a DA neuronal loss of ~32% and ~66% for nigral neurons and ~22% and ~46% for VTA DA neurons in these animals relative to those of the controls, respectively, at 5 and 15 weeks after transgene activation (Fig. 3h, i). Moreover, TH⁺ innervation in the striatum of these mice was severely reduced, confirming extensive DA neurodegeneration (Fig. 3c, e). Similarly, the number of nigral NeuN+ neurons was reduced to a similar extent, confirming progressive neuronal loss

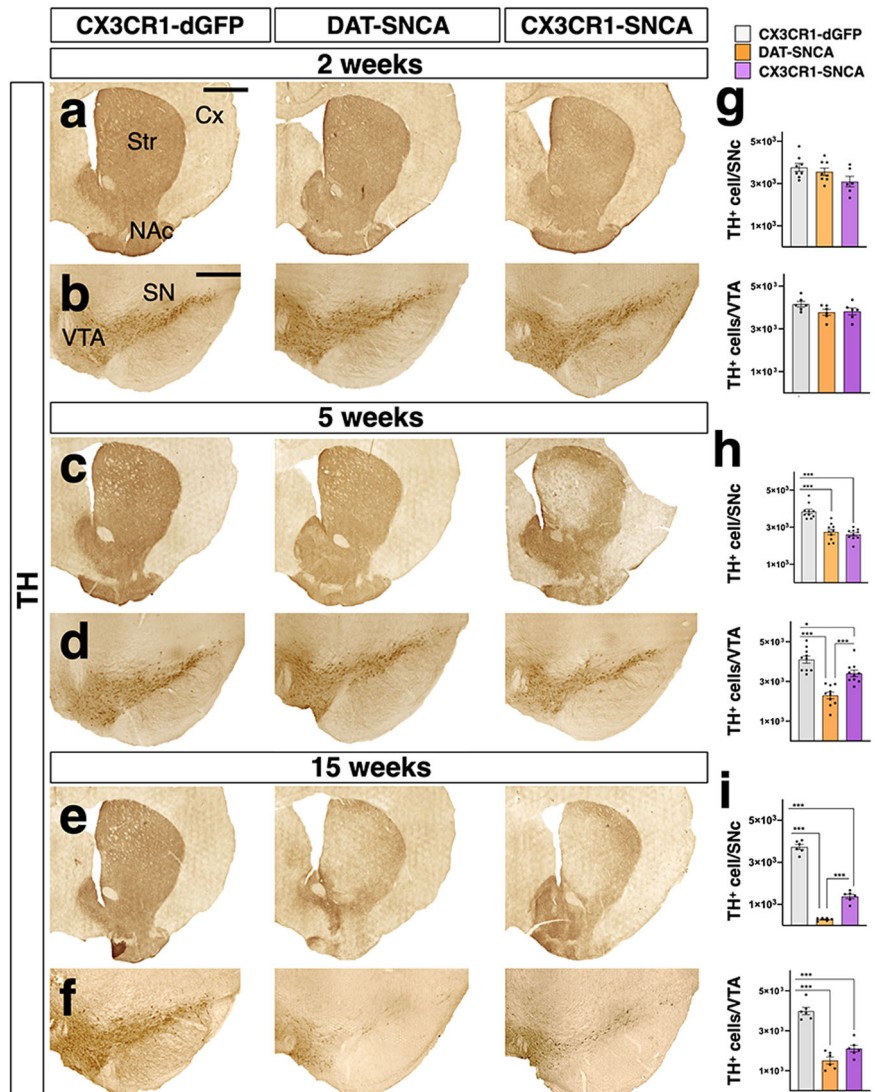

**Fig. 3 Microglia-specific αSYN accumulation leads to severe DA neuronal degeneration in CX3CR1-SNCA mice. a, c, e** TH immunostaining in striatum shows a progressive loss of DA fibers over time after viral αSYN expression in DAT-SNCA and CX3CR1-SNCA animals at 2, 5, and 15 weeks. **b, d, f–i** The unbiased stereological count of TH+ neurons in the nigral tissue reveals significant neurodegeneration from 5 weeks onwards causing a 28% ± 1.2% and 32% ± 0.7% cell loss in DAT-SNCA and CX3CR1-SNCA mice, respectively. The neurodegeneration is progressive in both models reaching 92% ± 0.3% and 66% ± 1.2% of TH+ cell loss after 15 weeks in DAT-SNCA and CX3CR1-SNCA mice, respectively. The number of TH+ neurons in the VTA decreases over time, causing a 17% ± 1.2 and 44% ± 1.2 neuronal loss in CX3CR1-SNCA and DAT-SNCA animals, respectively. After 15 weeks we observe a further reduction reaching 47% ± 1.5 and 62% ± 1.6 in CX3CR1-SNCA and DAT-SNCA mice, respectively. Data are the mean ± s.e.m from $n = 8$ (CX3CR1-dGFP, DAT-SNCA), $n = 6$ (CX3CR1-SNCA) (**g**), $n = 10$ (**h**), $n = 6$ (**i**) nigral tissues. One-way ANOVA is followed by Bonferroni's post-test. **g** Not significant differences for both SNc ($F_{2, 19} = 2.714$, $p = 0.09$) and VTA ($F_{2, 15} = 2.106$, $p = 0.15$). Panel **h** upper bar graph: ***$p < 0.0001$; $p > 0.9$ for (DAT-SNCA vs. CX3CR1-SNCA). Panel **h** lower graph: ***$p < 0.0001$ for (CX3CR1-dGFP vs. DAT-SNCA), ***$p = 0.0002$ for (DAT-SNCA vs. CX3CR1-SNCA) and *$p = 0.022$. **i** ***$p < 0.0001$. Scale bar, 500 μm. **a, c, e** 300 μm **b, d, f** The loss of TH positive cells is accompanied by the progressive reduction of TH positive fibers in the striatum of transduced mice. Cx cerebral cortex, NAc nucleus accumbens, SN *substantia nigra*, Str striatum, VTA ventral tegmental area.

and not simply a lack of TH expression (Supplementary Fig. 7a). Conversely, stereological counting revealed that the NeuN+ neuronal population in the *substantia nigra pars reticulata* was not affected in either experimental group 15 weeks after viral transduction (Supplementary Fig. 7b). These results indicate that microglial-specific αSYN accumulation triggers the loss of midbrain DA neurons through non-cell-autonomous mechanisms. Surprisingly, the extent of DA neuronal degeneration was similar both in numbers and temporal dynamics between treated DAT-SNCA and CX3CR1-SNCA mice, although the former group displayed greater cell loss after 15 weeks, suggesting that intrinsic

neuronal or microglial αSYN accumulation triggers strong detrimental effects on nigral DA neuronal survival. Accordingly, αSYN expression in both neuronal and glial cells mediated by injection of a lentiviral vector with αSYN downstream to a constitutive promoter in wild-type mice roughly doubled the levels of DA neuronal cell loss at 5 weeks (~66%) (Supplementary Fig. 8). Thus, diffuse αSYN viral expression induced an additive toxic effect, verifying that independent cell death mechanisms are triggered by αSYN accumulation in neurons and glial cells. Next, we generated a FLEX-LV for the conditional expression of the wild-type αSYN which was injected in CX3CR1-creERT2 mice for selective

expression in microglia. αSYN protein levels sustained by FLEX-LV with either SNCA wild-type or mutant (G420A) were comparable in nigral tissues (Supplementary Fig. 4d, e). Importantly, 5 weeks after viral delivery the treated mice exhibited a strong microgliosis associated with significant loss of nigral DA neurons (Supplementary Fig. 9). On contrary, wild-type αSYN expression in DA neurons in DAT-cre mice did not elicit frank microglia reactivity with only an incipient neurodegeneration at 5 weeks post-treatment (Supplementary Fig. 9). Thus, these observations confirmed that both wild-type and mutant αSYN expression in microglia trigger DA neuronal cell death and promote reactive microgliosis. Though, the extent of neurodegeneration was higher using the mutant αSYN (~24% and ~32% using wild-type or mutant αSYN, respectively), which then was employed for the following experiments.

**αSYN accumulation in microglia triggers a strong inflammatory reaction**. To determine the pathological processes initiated in αSYN-accumulating microglia that lead to neurodegeneration, we performed total RNA-seq analysis of isolated nigral tissue from DAT-SNCA, CX3CR1-SNCA. and control mice 5 weeks after transduction (Supplementary Fig. 10a). Interestingly, αSYN accumulation in microglial cells resulted in strong enrichment of genes encoding (i) critical members of the inflammasome pathway (*Nlrp1a, Nlrp3, Nlrc3, Pycard, Aim2, Casp1, Il1a,* and *Il18*), (ii) inflammatory cytokines (*Il1a, Il1b, Il18,* and *Ifng*) and (iii) essential components of inflammatory nuclear transcriptional pathways (*Irf1, Irf7, Irf9, Stat1, Stat3, Ikbke,* and *Nfkbie*) (Supplementary Fig. 10b, c). Increased levels of some of these genes in CX3CR1-SNCA tissues relative to either DAT-SNCA or control tissues were confirmed in independent samples by targeted qPCR analysis (Supplementary Fig. 10d). These data strongly pointed to a marked inflammatory state of CX3CR1-SNCA microglia. Accordingly, microglial cells in CX3CR1-SNCA tissues showed strong immunoreactivity for ASC, a protein that plays a pivotal role in inflammasome assembly and function (Supplementary Fig. 10e). In addition, CX3CR1-SNCA samples showed robust upregulation of a plethora of chemokines and their receptors, some of which are known to be specifically activated in inflammatory microglia (*Ccl2, Ccl3, Ccl4, Ccl7, Ccl12,* and *Cxcl10*) (Supplementary Fig. 10c). Taken together, these findings provide evidence that microglia exhibited the highest inflammation levels in CX3CR1-SNCA mice, and this effect was likely caused by the cell-autonomous accumulation of αSYN. Intriguingly, we noted that among the most enriched chemokines in CX3CR1-SNCA tissue were those known to attract T cells and peripheral monocytes (*Ccl1, Ccl2, Cxcl10,* and *Cxcl11*). Thus, we queried the bulk RNA-seq datasets for T cell-specific genes that were expressed in the nigral tissues of the three groups of animals. Consistent with our expectations, several T cell-specific genes were found to be strongly upregulated in CX3CR1-SNCA nigral tissue (*Cd3e, Cd8a, Cd4, Cd7,* and *Cd160*; Supplementary Fig. 11a). Thus, we examined the nigral tissue for the presence and localization of T cells by immunostaining for CD3. Consistent with the gene expression results, we identified a large number of CD3[+] T cells scattered within the nigral area in CX3CR1-SNCA mice and a smaller number of these cells in the DAT-SNCA nigrae; conversely, in control tissues, these cells were hardly represented (Supplementary Fig. 11b–d). CD3[+] T cells in CX3CR1-SNCA mice were negative for human αSYN immunostaining, again indicating the specificity of the genetic system and the absence of αSYN spreading from the targeted microglia (Supplementary Fig. 11e). Intrigued by these findings, we analyzed human autoptic brain nigral tissue and found that a large number of CD3[+] T cells clustered around vessels but were also

scattered in the parenchyma exclusively in PD patients and not in age-matched healthy donors (Supplementary Fig. 12). Taken together, these data verify the inflammatory state caused by αSYN accumulation in CX3CR1-SNCA microglia, activating proinflammatory cytokines and chemokines, which in turn attract peripheral immune cells within nigral tissue.

**Cellular composition of the innate and adaptive immune system in CX3CR1-SNCA and control nigral tissues**. We next aimed to decipher the diversity of immune cells in CX3CR1-SNCA and control nigral tissues by single-cell transcriptomic analysis. CD45[+] cells were isolated by fluorescence-activated cell sorting (FACS) from the dissected nigral tissues of CX3CR1-SNCA and control mice 5 weeks after lentiviral infection and directly processed for droplet-based sequencing. A robust increase in total CD45[+] cells was found in CX3CR1-SNCA tissues relative to control nigral tissue samples, which reflected an expansion of both CD11b[+] (myeloid cells) and CD11b[−] (lymphocytes) cellular subsets (Fig. 4a). A total of 3378 and 720 CD45[+] cells were sequenced with an average sequencing depth of 1554 transcripts per cell. Integrated automatic clustering between the CX3CR1-dGFP and CX3CR1-SNCA conditions identified 15 different populations (Fig. 4b). Microglia were distributed in four different cell populations (M-A1-4) expressing cardinal genes of this cell type (for example, *Csf1r, Cx3cr1, Tmem119,* and *P2ry12*) (Fig. 4b, c). CX3CR1-SNCA cells in M-A2/3/4 clusters displayed the highest expression of multiple inflammatory genes, including *Il1a, Il1b, Tnf, Itgax, Csf1, Clec7a, and Ptgs2* (Fig. 5a, b). In addition, these cells showed enriched expression of genes encoding glycolytic enzymes (*Ldhb* and *Pgk1*), complement components (i.e., *C1q*) (Fig. 5d), major histocompatibility complex (MHC) class II proteins (*B2m, H2-Aa* and *H2-DMa*), *Tlr2* and *Tlr4* (Fig. 5a, b, d). Interestingly, one of the most differentially expressed genes was *Apoe*, which has been recently shown to be strongly upregulated in microglia in other neurodegenerative disease models (Alzheimer's disease, ALS and MS) but not in PD[37]. Conversely, these cells showed suppression of most of the known homeostatic microglial genes (*Mef2a, Csf1r, Mertk, Tgfbr1,* and *Jun*) (Figs. 4c and 5b)[37,38]. Notably, although these clusters contained cells of both CX3CR1-dGFP and CX3CR1-SNCA conditions, control cells were grouped together for their low expression of markers of the activation state (Figs. 4b and 5b). Conversely, cluster M-A1 included microglial cells that maintained an overall homeostatic state gene signature with the highest representation of control cells that displayed a virtual absence of inflammatory transcripts (*Itgax, Cst7,* and *Apoe,* among others) (Fig. 5b). Thus, this cluster represented a microglial population with a less activated state and mostly represented among the control cells (Fig. 4b). Interestingly, human *SNCA* transgene expression was mainly found in the most highly activated microglial cell populations (M-A2/3/4), indicating a direct correlation between αSYN accumulation and microglial reactivity (Fig. 4c). The cell clustering did not highlight a separate population of peripheral monocytes, probably due to the high inflammatory condition in the CX3CR1-SNCA condition which further enhances the molecular similarity with the resident reactive microglia. Clusters 6, 7, and 8 were associated with CD8[+] T cells that shared molecular signature associated with an activated state and expression of the *Ifng* and granzyme (*Gzmb* and *Gzmk*) and cathepsin W genes (Figs. 4c and 5a, c, d). In particular, genes associated with cytotoxic (*Gzmb, Gzmk,* and *Prf1*) and memory effector (*Cxcr3, Cxcr6, Cd7,* and *Il7r* high/*Cd62l* and *Ccr7* low) T cells were associated with cluster CD8-1/2 mostly represented by CX3CR1-SNCA cells (Figs. 4b and 5c). Group CD8-2 was characterized by high levels of genes with roles in cell

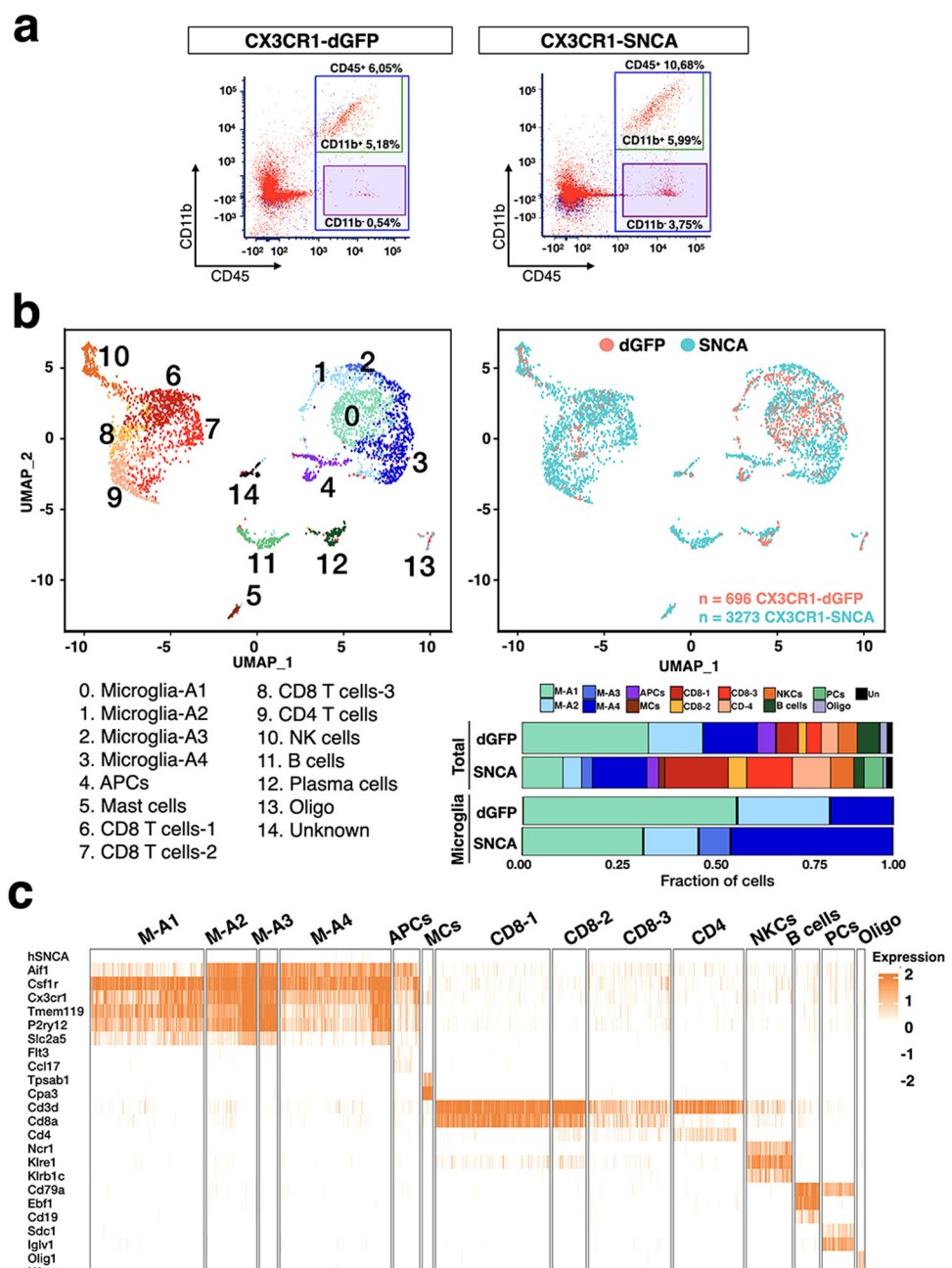

**Fig. 4 Single-cell transcriptomics of the brain immune cells 5 weeks after viral transduction. a** Flow cytometry-based isolation of nigral CD45+ cells in CX3CR1-Cre animals injected with lentiviral particles expressing dGFP (CX3CR1-dGFP) or *SNCA*^G420A (CX3CR1-SNCA). **b** Upper panels**:** Uniform Manifold Approximation and Projection (UMAP) clustering of the different populations of CD45+ cells based on gene expression profiling. Lower panel: bar plots highlighting the fraction of cells respect to total number of cells (up) or only the microglia group (down) for each of the clusters between the two different conditions (dGFP and SNCA). **c** Heatmaps showing representative cell-type specific genes defining unique cell populations and their respective clusters. Orange or white color code represents high or low gene expression levels, respectively. MCs mast cells, NKCs natural killer cells, PCs plasma cells, Oligo oligodendrocytes.

proliferation, distinguishing a population of cytotoxic CD8+ T cells in the rapid cell cycle (Fig. 5c). Conversely, the CD8-3 cell cluster appeared to include both naïve cells (high *Ccr7*, *Foxp1*) and low activation cells associated with low levels of the *Ifng* and *Gzmc* genes (Fig. 5a, c) and was the highest enriched for cells belonging to the CX3CR1-dGFP control condition (Fig. 4b). Cluster 9 was associated with CD4 Th1+ T cells with high expression levels of *Il2ra, Tnfrsf4, Ccr7, Bhlhe40,* and *Ifng* (Fig. 5a, c). Cluster 4, highly enriched in the CX3CR1-SNCA mice, was populated by *Ccl22*-expressing antigen-presenting cells (APCs),

largely represented by conventional cross-presenting dendritic cells expressing the typical molecular markers *Xcr1*, *Treml4*, and *Batf3* (Fig. 5a). NK cells were clustered into group 10 and distinguished by transcripts encoding KLR family members (*Klre1, Klra7,* and *Klrb1a*) and the transcription factor *Irf8* (Fig. 5c). Interestingly, these cells expressed high levels of *Ifng*, *Gzmb*, and *Prf1*, suggesting high potential cytotoxic activity. Clusters 11 and 12 were associated with mature B cells (for example, *Cd20, Cd19,* and the immunoglobulin genes *Iglc1* and *Iglc3*) and plasma cells (*Iglv1, Sdc1,* and *Slamf7*), respectively (Fig. 5c). Cluster 5 was

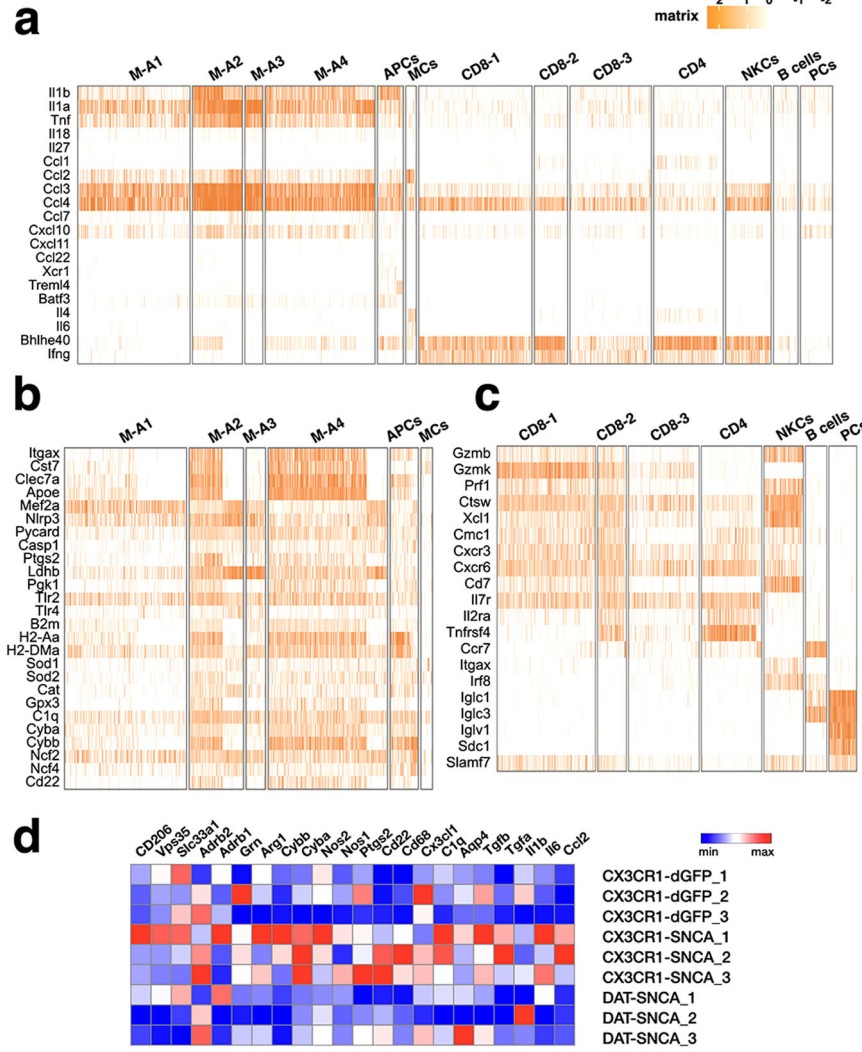

**Fig. 5 Single-cell expression pattern of genes encoding for distinctive cytokines, chemokines, and inflammation markers. a–c** Heatmaps showing the expression pattern of selected genes encoding for cytokines and chemokines (**a**), inflammation markers and oxidative molecules (**b**), and markers for different reactive states (**c**). MCs mast cells, NKCs natural killer cells, PCs plasma cells, Oligo oligodendrocytes. **d** qPCR analysis on independent biological samples for selected representative genes.

characterized by mast cells with high peptidase expression (for example, *Mcpt2*, *Cpa3*, and *Tpsab1*) (Fig. 4c) and the *Il4* and *Il6* cytokine-encoding genes and mostly represented by CX3CR1-SNCA cells (Figs. 4b and 5a).

In sum, immune cells in CX3CR1-dGFP control mice displayed a very limited activation state which might be the consequence of the mild tissue damage associated with brain injections, with few infiltrated cells. On the contrary, CX3CR1-SNCA mice presented a complex composition of brain immune cells, with numerous cell types of the peripheral adaptive immune system that exhibited an high inflamed state.

**Neurotoxicity of αSYN-accumulating microglia is mediated by phagocytic exhaustion with heightened ROS and NO production**. Given the high inflammatory state of microglia in CX3CR1-SNCA mice, we sought to investigate the cell-autonomous consequences of αSYN accumulation in this cell type. First, we noted that antioxidative genes were strongly upregulated (for example, *Sod1*, *Sod2*, and *Cat*), indicating an unbalanced redox state in CX3CR1-SNCA microglial cells. Similarly, *Nos2*, which encodes inducible nitric oxide synthase (iNOS), was highly expressed in CX3CR1-SNCA nigral tissues (Fig. 5d). In addition, *Cyba* and

*Cybb*, which encode the two catalytic subunits of the NADPH oxidase Nox2 (Gp91phox), and the *Ncf1*, *Ncf2*, and *Ncf4* genes, which produce the cytosolic subunits p40phox, p47phox, and p67phox, respectively, were robustly activated in these cells (Fig. 5b, d). Nox2 catalyzes the production of superoxide through electron transfer from cytosolic NAPDH to oxygen. Nox2 is primarily located on phagosomes or cellular membranes, where it is strategically positioned to produce superoxide within phagosomal vesicles or in the extracellular space. Increased Nox2 activity has been associated with several neurodegenerative diseases and induces high levels of oxidative stress, leading to high inflammation and neuronal cell loss[39]. However, we thought that alterations in Nox2 protein localization might also change the ratio of intracellular and extracellular ROS production, potentially leading to detrimental effects. Intriguingly, we found that *Cd22* was strongly upregulated in highly inflamed microglia in CX3CR1-SNCA nigral tissue (Fig. 5b, d). CD22 was recently described as a critical negative regulator of phagocytosis that led to exhausted microglia during aging[40]. Thus, phagocytic exhaustion and impaired phagosome formation might induce Nox2 protein relocation on the cell membrane, leading to excessive extracellular superoxide production. Intriguingly, we

detected several CD22/IBA1 double-positive cells in the autoptic nigral tissue of PD patients (Supplementary Fig. 13), suggesting that an abnormal increase in CD22 is a pathological feature that is conserved between CX3CR1-SNCA mice and humans with the disease.

To directly assess the link between αSYN accumulation and oxidative toxicity in CX3CR1-SNCA microglial cells, we established cultures of primary microglial cells from healthy mice and assessed their response to αSYN modulation. In particular, we initially examined whether αSYN intracellular accumulation by lentiviral-mediated gene upregulation or exposure to synthetic αSYN fibrils induced similar gene expression alterations. Thus, we generated synthetic human αSYN fibrils by in vitro aggregation of the recombinant protein and sonication as previously described[41,42]. In addition, we included an additional condition with recombinant IFNγ to not ignore the effect of this inflammatory cytokine, which is highly produced by infiltrated T and NK cells in CX3CR1-SNCA mice. Of note, αSYN fibrils and lentiviral transduction triggered very similar changes in the expression of genes in this analysis, suggesting that these processes were similarly altered even when intracellular αSYN accumulation was achieved through αSYN fibril-induced seeding (Fig. 6a). This result also highlighted that IFNγ potentiated the effects of αSYN, and only when both were present did the tested genes show the highest expression changes (Fig. 6a). Thus, these findings suggested that IFNγ and αSYN acted synergistically to stimulate the expression of oxidative genes and *Cd22* in microglial cells. Next, we examined whether αSYN fibrils or LV-mediated SNCA$^{G420A}$ expression in the presence or absence of IFNγ could affect phagocytic activity in primary microglial cells. Initially, we found that αSYN fibrils caused impaired autophagy-mediated protein degradation in microglial cells, as demonstrated by the diffuse accumulation of P62 aggregates in microglial cells, and this effect was further exacerbated by concomitant IFNγ exposure (Fig. 6b, d). To directly test phagocytic activity, we incubated αSYN- and IFNγ-treated microglia with brain-derived synaptosomes loaded with pHrodo, a fluorescent indicator of cellular uptake to acidic compartments and lysosomes. Importantly, microglia exposed to Atto488‐conjugated αSYN fibrils or treated with LV-SNCA$^{G420A}$ showed dramatic loss of synaptosome uptake, which was further amplified by exposure to IFNγ (Fig. 6c, e). Thus, *Cd22* gene upregulation correlated with the loss of phagocytotic activity and the generation of exhausted microglia. We next attempted to measure the levels of oxidative stress after αSYN fibril and IFNγ treatments. Consistent with the previously described transcriptional changes, incubation with αSYN fibrils and IFNγ led to a strong increase in the levels of NO released by microglial cells into the culture media (Fig. 6e). Moreover, we used ROS-sensitive dichlorofluorescein (H$_2$DCFDA) dye to monitor intracellular ROS levels. Importantly, the H$_2$DCFDA signal was upregulated after exposure to αSYN fibrils and was further enhanced by the presence of IFNγ (Fig. 6g). Finally, we examined whether the high inflammatory status of αSYN-accumulating microglia could create a toxic environment that could drive neuronal loss. Thus, fresh culture media from control and treated microglia were administered to primary DA neuronal cultures established from E13.5 mouse embryonic nigral tissue (Fig. 7). The medium from microglia was maintained for 1 week on the neuronal cultures before the total number of DA neurons was assessed. Medium from untreated microglia was not detrimental, since the percentage of DA neurons was not different from that of neuronal cultures maintained in their specific culture conditions. In contrast, treated microglia-conditioned media induced significant degeneration of DA neurons, and the highest loss was associated with medium from microglia exposed to both αSYN

and IFNγ. In accordance with these cell culture data, we found that the oxidative stress marker 4-HNE was upregulated in CX3CR1-SNCA DA neurons at unexpectedly higher levels than in analogous DAT-SNCA neurons (Supplementary Fig. 14). Collectively, these observations indicate that αSYN-accumulating microglia developed strong phagocytic impairment that led to severe exhaustion, which correlated with excessive oxidative and nitrosative levels that induced DA neuronal loss both in vitro and in vivo.

**Inhibition of ROS or NO toxicity reduces neurodegeneration in CX3CR1-SNCA mice.** We next sought to obtain proof-of-principle that limiting the detrimental effects of inflamed microglia could restrain neurodegeneration in CX3CR1-SNCA mice. First, we used minocycline, a synthetic tetracycline derivative that has multifaceted functions and is capable of attenuating microglial activation[43] and inhibiting inflammasome-mediated Casp1 activation[44]. CX3CR1-creERT2 mice were administered minocycline once per day intraperitoneally (50 mg/kg) soon after viral transduction and tamoxifen-induced transgene activation. After 5 weeks of treatment, the number of nigral DA neurons was assessed by TH immunohistochemistry and unbiased stereology. Treated CX3CR1-SNCA mice showed significant loss of TH + neurons at 5 weeks, and this effect was significantly attenuated by chronic minocycline administration (Fig. 8a). Consistent with the known anti-inflammatory effect of minocycline, there was a marked reduction in hypertrophic microglia, as indicated by reduced IBA1+ signal intensity in the nigral tissue of CX3CR1-SNCA animals that received drug treatment (Fig. 8a). Next, given the high levels of nitrosative stress induced by αSYN accumulation in microglia, we treated CX3CR1-SNCA mice with 1400 W, a brain-penetrating and highly selective inhibitor of iNOS. Analogous to minocycline, 1400 W was administered to CX3CR1-SNCA mice by daily IP injections (10 mg/kg) soon after viral transduction and tamoxifen-induced transgene activation. After 5 weeks, 1400 W-treated mice had significantly more surviving TH + DA neurons in nigral tissue than PBS-injected controls, as assessed by stereological counting after TH immunohistochemistry (Fig. 8b). These results collectively indicate that compounds that reduce ROS and NO toxicity can restrain the progressive loss of DA neurons in CX3CR1-SNCA mice, at least when administered soon after SNCA transgene induction with a chronic modality.

**Discussion**
In this study, we developed a transduction viral protocol to selectively express the human *SNCA* mutant gene in microglial cells or DA neurons through conditional expression in CX3CR1-CreERT2 and DAT-Cre mice, respectively. In CX3CR1-SNCA animals, lentiviral-mediated transgene activation was restricted to microglia with no detectable residual expression in other neuronal or glial cell types. AAV particles are not suitable for this application, since no AAV serotype or engineered variant can efficiently promote transgene expression in brain microglial cells in vivo. Moreover, we reported that AAV-FLEX vectors were not sufficient to induce sufficiently selective expression in DAT-Cre mice, as indicated by spurious transgene expression in unrelated cell types, raising a serious concern for the use of these viruses for cell type-specific gene misexpression in vivo, which corroborated recent similar findings[34]. Here, we unexpectedly found that CX3CR1-SNCA mice developed progressive and selective DA neuronal loss. Importantly, although pathological αSYN aggregation was detectable in microglia in these mice, DA neurons did not show any evidence of αSYN accumulation, indicating that DA neuronal degeneration was largely independent of this pathological mechanism. On the other hand, bulk and single-cell

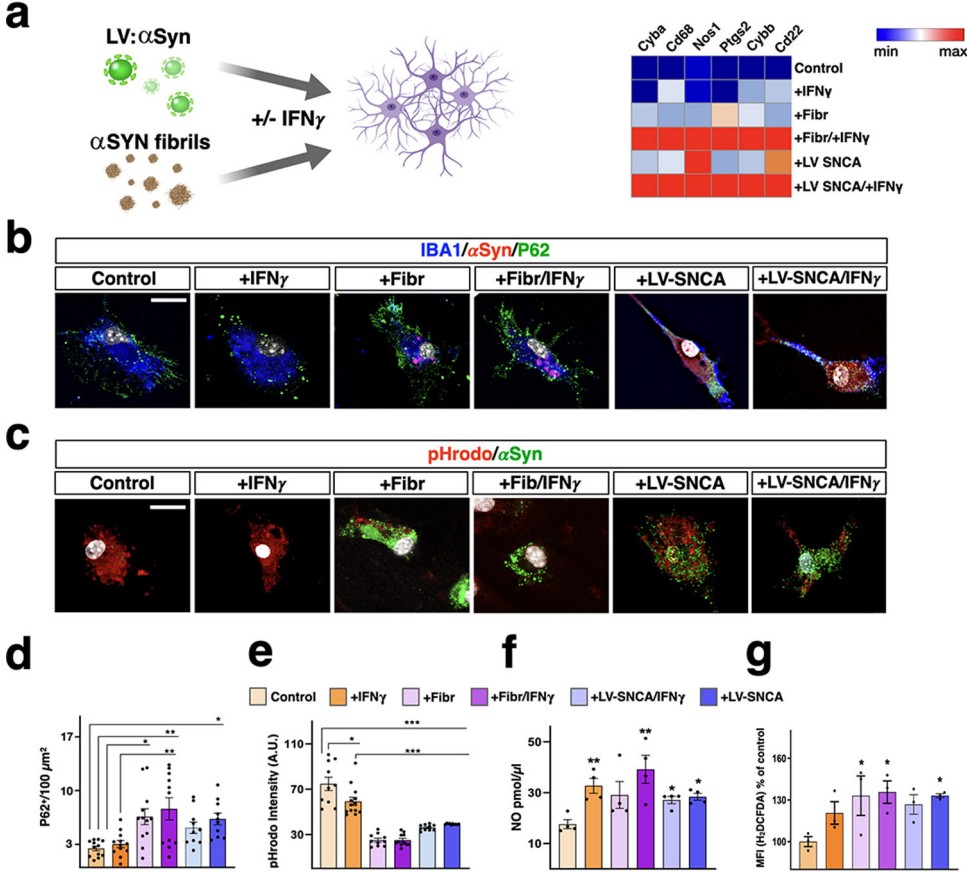

**Fig. 6 αSYN and IFNγ provoke phagocytosis exhaustion in microglia. a** Either lentiviral αSYN gene expression or pre-formed αSYN fibrils elicit similar expression changes in key redox genes in primary microglia. **b, d** P62 Immunostaining (green) in microglia exposed to exogenous αSYN fibrils or expressing exogenous αSYN (red) with or without, IFNγ. Both αSYN fibrils and αSYN overexpression in presence of IFNγ provoke the accumulation of P62$^{+}$ vesicles compared to control ±IFNγ. Data are the mean ± s.e.m. from $n = 12$ cells from two different experiments. *$p = 0.013$ (control vs. +Fibr), $p = 0.043$ (control vs. +LV-SNCA/ + IFNγ); **$p = 0.001$ (control vs. +IFNγ/+Fibr), $p = 0.005$ (+IFNγ vs. +INFγ/+Fibr). **c, e** Assessment of microglial phagocytosis by measuring the internalization of pHrodo labeled synaptosomes (red). The phagocytosis rate is scored by quantification of the intracellular pHrodo fluorescence intensity. Phagocytosis is impaired in microglia treated with mouse IFNγ and worsened in the presence of αSYN fibrils or αSYN overexpression (green). Data are the mean ± s.e.m from two independent experiments. ***$p < 0.0001$, *$p = 0.007$. Scale bar, 15 μm. **f** Nitric oxide (NO) extracellular release by microglia is increased by IFNγ, αSYN fibrils, and αSYN overexpression and further enhanced by αSYN fibrils and INFγ co-application. The extracellular NO level is assessed by fluorimetric analysis on the cell culture medium. Data are the mean ± s.e.m from $n = 4$ independent experiments. *$p = 0.045$ (control vs. +LV-SNCA), $p = 0.024$ (control vs. +LV-SNCA/ +IFNγ) and **$p = 0.006$ (control vs. +IFNγ), $p = 0.001$ (control vs. +IFNγ/+Fibr) **g** Intracellular ROS significantly increase in microglia exposed to αSYN fibrils with a further surge when IFNγ is co-applied. In αSYN overexpressing microglia, ROS levels reach a significant increase only in presence of IFNγ. Data are the mean ± s.e.m from $n = 3$ independent experiments. *$p = 0.02$ (control vs. +Fibr), $p = 0.011$ (control vs. +IFNγ/+Fibr) and $p = 0.047$ (control vs. +LV-SNCA). For detailed statistics see Supplementary Data 1.

transcriptomics analysis defined the molecular profiles of αSYN-accumulating microglia, confirming their strong inflammatory status and high expression of genes encoding proinflammatory cytokines and chemokines, complement and MHC-II components, inflammasome-related proteins, and TLR receptors. Interestingly, among the most upregulated genes in αSYN-accumulating microglia, we identified *Apoe*, high levels of which are a feature of microglia in models of Alzheimer's disease, multiple sclerosis, and amyotrophic lateral sclerosis[37]. Since APOE ε4 was shown to directly exacerbate αSYN pathology and reactive gliosis in PD mice[45], it is plausible that a vicious loop between inflamed microglia and the APOE genotype can strongly contribute to neurodegeneration. CD22 was another gene with strong upregulation in αSYN-activated microglia. A recent study by Pluvinage and colleagues described CD22 upregulation during aging and its inhibitory effect on microglial phagocytosis[3]. Consistent with these findings, αSYN accumulation in microglia led to severe loss of phagocytic activity with consequent P62 protein

accumulation. This process is likely conserved in human pathology, since we showed that numerous CD22$^{+}$ cells were found in autoptic PD but not control brain nigral tissues in the proximity of DA neurons. Hence, long-term CD22 blockade by antibody targeting or genetic means might be a valuable approach to restore microglial functionality in CX3CR1-SNCA mice and possibly in humans. Microglia in CX3CR1-SNCA mice displayed high levels of oxidative stress-related genes, suggesting a marked imbalance in the intracellular redox state. Furthermore, these cells exhibited excessive transcriptional levels of genes encoding the catalytic and regulatory subunits of the NADPH oxidase Nox2. Considering that this enzyme is exclusively intercalated in the phagosome or cell membrane, exhausted microglia lacking phagosomes will displace most of this enzyme in the plasmalemma. Thus, considering the prevalence of this enzyme and its cell membrane enrichment, exhausted microglia might become a powerful source for the production of extracellular ROS, leading to the generation of a local toxic environment. A further

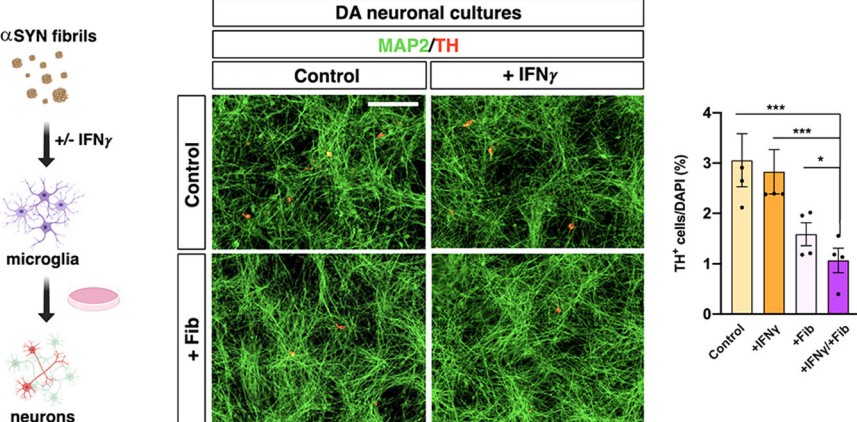

**Fig. 7 αSYN and IFNγ induce toxic microglia.** Culture media of microglia cells exposed to either IFNγ or αSYN elicit a significant loss of DA neurons (red) which is synergically increase when both treatments are combined. Data are the mean ± s.e.m from n = 4 independent experiments. Two-sided Kruskal–Wallis test is followed by the two-stage linear step-up procedure of Benjamini, Krieger, and Yekutieli. ***p = 0.003 (control vs. +IFNγ/+Fibr), p = 0.009 (+IFNγ vs. +Fibr/+IFNγ) and *p = 0.03 (+Fibr vs. +Fibr/+IFNγ). Scale bar, 500 μm.

mechanism contributing to this effect is the enhanced generation of NO sustained by the increased expression of the nitric oxide synthase-encoding gene *Nos2*. Importantly, due to its reactivity with ROS, the concomitant production of NO will likely stimulate the production of reactive nitrogen species, such as peroxynitrite (ONO−), which is one of the most potent oxidant molecules released by living cells[46]. Consistent with the changes in transcriptional gene expression, we found that αSYN accumulation in microglia led to enhanced intracellular ROS levels, increased NO production, and impaired glutathione homeostasis. The considerable impact of these toxic species on DA neuronal survival was further demonstrated by the beneficial effects of compounds that inhibit ROS and NO production on treated CX3CR1-SNCA mice. Pharmacological treatment was initiated concomitantly with tamoxifen-induced expression of the lentiviral transgene and maintained for the entire period before analysis. Given this administration schedule, these treatments were effective in preventing disease progression but not reversing existing pathology. Thus, these treatments should be considered more as proof-of-principle strategies to demonstrate the detrimental effects of these mechanisms rather than approaches with direct translational relevance. However, since inflammation is increasingly recognized as occurring early in PD development and contributes to disease progression in the human pathology, targeting inflammatory pathways is predicted to be effective only if started sufficiently early during pathogenesis and before frank degeneration is achieved. Therefore, the success of these therapeutic approaches is likely dependent on improvements in early clinical diagnosis. Herein, we also provided the molecular description of brain resident and peripheral immune cells in mice with ongoing DA neuronal degeneration. In particular, CX3CR1-SNCA nigral tissue was enriched in numerous populations of innate and adaptive immune cells that originated in the periphery and infiltrated during the neurodegenerative process. A large fraction of this infiltrate was represented by cytotoxic CD8+ and CD4 Th1+ T cells and NK cells that shared high expression of IFNγ. This proinflammatory cytokine can engage microglial cells and induce changes in microglial activation[47,48]. Herein, we demonstrated that microglia exposed to both αSYN and IFNγ developed an exacerbated inflammatory state with high expression levels of critical oxidative genes and maximal neuronal cytotoxicity (Figs. 6 and 7). These findings are in accordance with previous observations describing the strong synergistic effect between IFNγ and LPS to drive microglial activation and toxicity[49]. Intriguingly, TLR4 is the receptor that binds to LPS, which activates the NF-kB intracellular pathway to modulate gene expression and induce microglial activation. Similarly,

extracellular aggregated αSYN can interact with TLR2 and activate similar intracellular signaling, leading to reactive microglia[29]. Thus, IFNγ can exacerbate the inflammatory response initiated by αSYN in microglia, promoting a feed-forward vicious cycle that reinforces the local inflammatory response and leads to both increased recruitment of peripheral leukocytes and increased microglial neurotoxicity. On the other hand, given the highly activated state of leukocytes in CX3CR1-SNCA mice, it is plausible that these cells might directly damage the survival of DA neurons in nigral tissue. Moreover, our transcriptome analysis revealed the infiltration of APCs, mast, and plasma cells, which can further fuel neurotoxicity and mediate neuronal cell loss (Fig. 8c). In particular, the presence of cross-presenting APCs might facilitate the immune response to cell-associated antigens[50]. This process would in principle facilitate the generation of autologous T cell reactivity against αSYN-derived epitopes, which has been recently described in PD patients[51,52]. CX3CR1-SNCA mice are not a comprehensive model for PD since αSYN accumulation in microglia is not a first etiopathological event in the human condition. In addition, in the human disease microglia accumulate mainly extracellular αSYN released from neurons. However, herein we forced SNCA expression in microglial cells in order to accelerate αSYN accumulation and exacerbate its dependent microglial response. Thus, these animals represent a unique system in which the responses by either the neurons or the adaptive immune system exerted to the αSYN intoxicated microglia can be selectively investigated to decipher the underlying disease mechanisms and identify and validate new immune-mediated translational approaches. Herein, we unveiled the neurotoxic effects of microglia accumulating αSYN, but not native GFP or its destabilized form. These findings raise the intriguing possibility that, beyond αSYN, other aggregation-prone proteins, like TAU and TDP-43, might share a similar neuropathological mechanism contributing to the progression of other relevant neurodegenerative diseases.

## Methods

**Viral vectors and lentiviruses production**. Two distinct sets of lox cassettes (lox2272 and loxP), interposed by XbaI site, were cloned downstream of the Ef1a promoter in a third-generation lentiviral backbone. GFP and *SNCA*[G420A] coding sequence were cloned in XbaI in antisense configuration. The FLEX-GFP and FLEX-*SNCA*[G420A] cassettes were then removed and cloned in AAV backbone under the CBA promoter.

Lentiviral replication-incompetent, VSVg-coated lentiviral particles were packaged in 293T cells (ATCC, Cat# CRL-3216)[53]. Cells were transfected with 30 μg of vector and packaging constructs, according to a conventional CaCl₂ transfection protocol. After 30 h, medium was collected, filtered through 0.44 μm

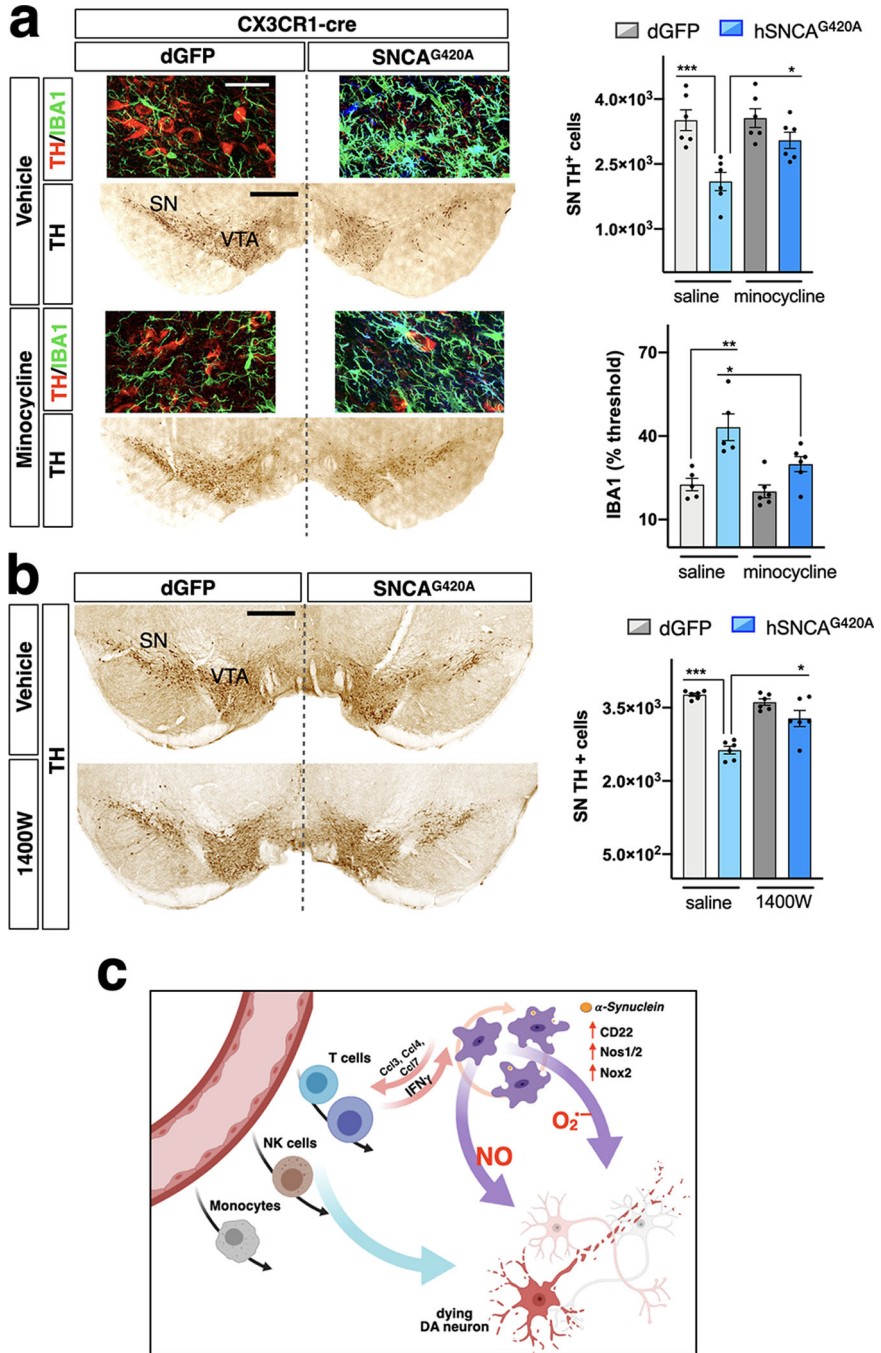

**Fig. 8 Chronic administration of minocycline and 1400 W attenuates neurodegeneration in CX3CR1-SNCA mice. a** Chronic minocycline administration significantly enhances survival of nigral DA neurons in CX3CR1-SNCA mice 5 weeks after viral transduction. αSYN microglial accumulation accounts for 40% ± 1.8% of DA neuronal cell loss compared to the control side, which is significantly rescued by the minocycline treatment (**a**, upper bar graph). The cell loss is accompanied by the ~2-fold increase of IBA1 signal compared to control side, which is attenuated by the minocycline treatment (**a**, lower bar graph). Data are the mean ± s.e.m from $n = 6$ nigral tissues. Two-way ANOVA is followed by Bonferroni's post-test. Upper bar graph: ***$p = 0.0009$, *$p = 0.03$; lower bar graph: **$p = 0.001$, *$p = 0.04$. Scale bar, 45 μm (fluorescence micrographs), 300 μm (IHC micrographs). **b** CX3CR1-SNCA mice treated with the iNOS inhibitor 1400 W showed a strong recovery in DA neuronal survival 5 weeks after viral transduction. Microglial expression of αSYN accounts for 28% ± 0.45% of dopaminergic cell loss compared to the control side, which is rescued by the 1400 W administration. Data are the mean ± s.e.m from $n = 6$ nigral samples. Two-way ANOVA is followed by Bonferroni's post-test. ***$p < 0.001$, *$p = 0.014$. Scale bar, 300 μm. **c** Illustration of the mechanisms underlying DA neuronal degeneration by αSYN accumulation in microglia. αSYN-dependent exhausted microglia upregulate chemotactic molecules to recruit peripheral immune cells. Infiltrated immune cells and αSYN accumulation synergize in establishing a vicious circle that fuel massive release of oxidant molecules and subsequent DA neuronal cell loss. SN *substantia nigra*, VTA ventral tegmental area.

cellulose acetate and centrifuged at 50,000×g for 2 h at 20 °C in order to concentrate the virus. AAV particles were produced as previously described[54].

**Animals and surgery procedures**. 9-weeks-old male Slc6a3tm1(Cre)Xz/J (DAT), Cx3cr1tm2.1(Cre/ERT2)Litt/WganJ (CX3CR1) and wild-type C57BL6 mice were purchased from Jackson Laboratories, housed under 12 h dark-light cycle, with a relative humidity of 50–60%, a temperature of 25 °C and fed ad libitum. Mice were bilaterally injected with LV-FLEX-$SNCA^{G420A}$, in DAT-Cre and CX3CR1-CreERT2 mice at a concentration of 1 X 10E9 vg/ml. Briefly, under isoflurane anesthesia, animals were injected with 1.5 μL of virus into the *substantia nigra pars compacta* according to the following coordinates: AP = −2.9/L = ±1.2, DV = −4.4 from skull. After the surgery, the CX3CR1-CreERT2 mice were intraperitoneally injected for 4 consecutive days with Tamoxifen (75 mg/kg in 1% ethanol/99% corn oil). The animals were then killed after 2, 5, and 15 weeks, the brains collected and subjected to the different procedures based on the type of analysis. All experiments were performed in accordance with protocols approved by internal Institutional Animal Care and Use Committee (IACUC 1142, Ospedale San Raffaele, Milan, Italy) and reported to Italian Ministry of Health according to the European Commission Council Directive 2010/63/EU. All quantifications were made by experimenters blinded to the experimental conditions.

**Chemicals and pharmacological treatment**. Minocycline (M9511 Sigma-Aldrich) was dissolved in a saline solution (1% DMSO) and daily injected intraperitoneally for 5 weeks at the dose of 50 mg/kg in a volume of 200 μL. 1400 W (W4262 Sigma-Aldrich) was dissolved in distilled water and intraperitoneally injected at the dose of 10 mg/kg for 5 weeks in a volume of 100 μL.

**Immunostaining analysis**. Brains were fixed in 4% of PFA, flash frozen, and cut into 50 μm slices. Next, the slices were permeabilized with 3% $H_2O_2$, 10% methanol, and 2% of triton for 20 m. After three washes the slices were treated with the blocking solution (3% BSA, 0,1% tween 20) and incubated with the primary antibody over night at 4 °C. For the immunofluorescence the secondary antibody was added, and the slices mounted for the imaging. When immunohistochemistry was preferable, the biotinylated secondary antibody was coupled with avidin/biotinylated enzyme complex for 1 h and then revealed with the DAB solution (Vector Laboratories). For those antibodies whose function depends on antigen retrieval, the slices were treated with the antigen retrieval solution (1 mM EDTA, 0.05% Tween 20, pH 8.0) and left at 80 °C for 30 min. All the antibodies used for this study are listed in Supplementary Table 1.

**Stereological cell count**. The unbiased stereological sampling method was used to quantify dopaminergic neurons in the nigral tissue. The cell counting was performed by using Leica DM400B motorized microscope equipped with Stereo Investigator software (MBF Bioscience). TH and NeuN positive cells in nigral tissue were counted at ×40 magnification over five 50-μm-thick sections collected every 200 μm, encompassing the whole *substantia nigra*. The optical fractionator stereological probe was then used to estimate the total number of TH and NeuN positive neurons.

**Acquisition and quantification of immunofluorescent images**. The immunofluorescence images were acquired with Leica TCS SP8 confocal microscope by using ×40 magnification lens. For the broad area images acquisition, MAVIG RS-G4 confocal microscope equipped with ×20 magnification lens was used. All the images were processed by using ImageJ software. To assure the unbiased quantification of the fluorescence images we blinded-assigned a fixed signal threshold for each experimental setup thus, we generated a macro for the automated image analysis. For the colocalization measurements we calculated the Mander's coefficient with JACoP image J plugin. The degree of the overlapping signals has been determined by using a fixed threshold values for both channels. The resulting Mander's coefficient generates a range of values from 1, a perfect correlation, to 0 representing a random distribution. With regards to the antibodies used in this work, their respective dilution and the catalog numbers see Supplementary Table 1 in supplementary methods section.

**Bulk RNA sequencing**. For bulk RNA sequencing the fresh brain tissue was collected from saline-perfused animals. Once the brain was removed, the nigral tissue was manually dissected and lysated in Trizol for the RNA isolation. RNA libraries were generated starting from 1 μg of total RNA whose quality was assessed by using a Tape Station instrument (Agilent). To avoid over-representation of 3′ ends, only high-quality RNA with an RNA Integrity Number (RIN) ≥ 9 was used. RNA was processed using the TruSeq Stranded mRNA Library Prep Kit. The libraries were sequenced on Illumina HiSeq 3000 with 76 bp stranded reads using Illumina TruSeq technology. Image processing and base calling were performed using the Illumina Real-Time Analysis Software. Fastq files were adapter-trimmed and quality-filtered with Trimmomatic. High-quality fastq files were aligned to the mouse genome by using STAR. Gene read counts were calculated with feature-Counts. Differential gene expression was performed with DESeq2. Downstream analysis was performed within the R environment.

**Real-time PCR**. Total RNA was isolated from primary microglia and animal tissues using the Qiagen RNeasy mini kit (QIAGEN). 1 μg of RNA was reverse transcribed with random hexamers as primers using ImProm-II™ Reverse Transcription System (Promega). For quantitative real time PCR (qRT-PCR), Titan HotTaq EvaGreen qPCR mix (BioAtlas) was used and expression levels were normalized respect to 18S expression. The results were reported as the $2^{-\Delta Ct}$. Primers for gene expression analysis are listed in Supplementary Table 2.

**Single-cell RNA sequencing**. The brain nigral regions were carefully dissected from two subsequent 500-μm-thick slices and the tissue directly subjected for RNA extraction or processed for the single-cell dissociation. The dissociation was performed using Adult Brain Dissociation Kit and OctoMACS Separator (both from Miltenyi Biotec) following manufacturer's instructions. This procedure collected a large fraction of different cell populations ranging from immune cells to neurons within the same preparation. Cells were stained with 1:200 Zombie Aqua Fixable Viability Kit (Biolegend) and subsequently with anti-CD45 PerCP-Cy5.5 antibody and anti-CD11b eFluor450 antibody (all from eBioscience). Cells were sorted using a FACSAria Fusion (BD) and collected in PBS 0.04% BSA to be processed for sequencing through the Chromium platform (10X Genomics). Cells were separated into droplet emulsion using the Chromium Single Cell 3′ Solution (V3.0) and Single-cell RNA-seq libraries were prepared according to the Single Cell 3′ Reagent Kits User Guide (V3.0). Libraries were sequenced on a Novaseq 6000 flowcell (Illumina), with a minimum depth of 50 K reads/cell.

We processed raw paired-end scRNA-seq data using cellRanger (10x Genomics) with default parameters to generate the DGE matrices[55]. We performed alignment to the mm10 reference genome using cellRanger "count" with Gencode v37 annotations. We achieved unique mapping for around 90% of the reads both in dGFP and SNCA conditions. We discarded non-uniquely mapped reads. To distinguish between those captured cellular transcriptions from those that captured ambient RNA, we sorted barcodes by decreasing number of reads and picked the inflection point ('knee') of the cumulative fraction of reads plot.

We used Seurat 4 for downstream computational analyses[56]. The two datasets were integrated first finding the anchors shared between them using the "FindIntegrationAnchors" function followed by IntegrateData merging. To remove damaged cells, we extracted the percentage of mitochondrial reads and the count of captured transcripts (nCount_RNA) and removed all barcodes with <200 nCount_RNA or high-percentage of mitochondrial reads (>20% in both d-GFP and SNCA). We removed barcodes with low mitochondrial reads (<0.8%) to exclude nuclei. In order to exclude potential doublets, we also excluded cells with very high nUMI (>5000). We were able to analyze 696 and 3273 cells for dGFP and SNCA conditions, respectively. For each cell, UMI counts per gene were normalized and scaled. We performed clustering considering only the top 2000 highly variable genes, as identified by the function "FindVariableGenes". Variable genes were then used to perform principal component (PC) analysis. We selected the PCs to be used for downstream analyses by evaluating the "PCEl-bowPlot" and the "JackStrawPlot". We used the first 20 PCs for both dGFP and SNCA conditions. We identified clusters using the function "FindClusters", which exploits a SNN modularity optimization clustering algorithm (at Resolution = 0.5). We visualized clusters using the uniform manifold approximation and projection (UMAP) dimensionality reduction. We used the manual inspection of marker genes determined using the "FindAllMarkers" function for cluster identification. This function determines which genes, that are expressed in at least three cells, are enriched in every clustering using $log_2FC$ threshold values of 0.25 and 0.05 of adjusted pValue (FDR).

**Primary cell cultures**. Mesencephalic primary neurons were established from E13.5 mouse embryos. The mesencephalic tissue was carefully dissected in HBSS medium and incubated with papain (9.7 U/mg in HBSS with EDTA 0.5 mM and L-cysteine 1 mM) for 30 min at 37 °C, washed with HBSS, and gently disaggregated with a 1 mL pipette tip. The cells were then resuspended in neurobasal medium solution (added with B27) and plated in a poly-L-lysine-coated 24 multi-well plate at the concentration of 400,000 cells/well. To obtain microglia primary culture, cortices of wild-type newborn mice were dissected and incubated at 37 °C for 30 m in 0.05% Trypsin EDTA (Merck) and 100 μg/ml DNAse I (Roche). Tissues were dissociated into a cell suspension and passed through a 40 μM strainer. After centrifugation at 550 × g for 10 m, cells from two mouse newborns were plated in 1 T-75 Matrigel GFR (Growth Factor Reduced, Corning) coated-flask to generate mixed glial cultures. The flasks were incubated at 37 °C in 5% CO2 in DMEM High Glucose supplemented with 10% FBS, 1X P/S, 1X Glutamine, 1X NEAA, and 1X Sodium Pyruvate (all from Merck) for 2–3 days without disruption. When confluency was achieved (after ~7 days), flasks were shaken at 350 rpm for 4 h at 37 °C to separate microglia from the astrocyte layer. Medium with microglia was collected and seeded in the desired plates after a Matrigel GFR coating. After a couple of days medium was changed and αSYN fibrils were added 15 ng/cm², as previously described (Volpicelli-Daley, 2014). After 7 days medium was replaced adding 100 ng/ml mouse IFNγ and procedure was repeated every other day for a total treatment of 4 days. At the end, cells were processed for subsequent analyses.

**Nitric oxide assay**. Nitric oxide in the microglia supernatants was detected using Nitric Oxide Assay Fluorometric Kit (Thermo Fischer Scientific) following manufacturer's instructions. Fluorescence was detected after 4 h by VICTOR3 Multi-label Plate Reader (PerkinElmer) and nitric oxide concentration for each point was calculated subtracting the value derived from the blank and extrapolating it from the total nitrite/nitrate standard curve.

**DCF assay**. Microglia was treated for 15 m with $H_2DCFDA$ (DCF, Invitrogen) 1:5000 diluted in PBS. Then cells were detached using Accutase Solution (Merck) 10 m at 37 °C. Data were acquired using a FACSCantoII (BD Bioscience) and were analyzed using FCS Express 7 (De Novo Software). Unstained samples were employed to set gates.

**Assay of microglial phagocytosis with synaptosomes**. Synaptosomes were isolated from the whole brain tissue by using the Syn-PER reagent (Thermo Fisher). Briefly, half of the brain was isolated and disaggregated into a glass Dounce homogenizer filled up with 4 mL of Syn-PER reagent. After 15 gently strokes the homogenate was poured in a 15 mL falcon and centrifuged at $1200 \times g$ for 10 m at 4 °C. The supernatant was then centrifuged at $15,000 \times g$ for 20 m at 4 °C. The pellet containing the purified synaptosome fraction was weighted for subsequent treatments. pHrodo Red SE (Thermo Fischer Scientific) was reconstituted in 150 ul DMSO (Merck) and added to synaptosomes resuspended in 0.1 M sodium carbonate (Merck) at a final concentration of 200 mg/ml. After 2 h of incubation on the shaker at 50 rpm in the dark, 2 PBS washes were performed and labeled synaptosomes were added to plates 0.5 mg/cm$^2$. After 3 h of incubation at 37 °C 5% CO2, four washes in PBS were performed and microglia was fixed by 4% PFA, counterstained with DAPI and analyzed with confocal microscopy (Leica TCS SP8).

**Fractal analysis of microglia morphology**. The features of microglia morphology were elaborated by FracLac tool of Image J software. Briefly, the immunofluorescent staining for IBA1 was acquired with 40x magnification lens and each single microglial cell was isolated from the context. The resulting image (containing one cell) was submitted to FracLac for the automatic detection of the Convex Hull and the Bounding Circle edges, together with the number of total and foreground pixels. The information was integrated to provide different indices recapitulating the cell shape, including the fractal dimension which represents a statistical index of the shape complexity and of the space-filling capacity of a pattern.

**Statistics**. Values are expressed as mean ± s.e.m. as indicated. All the statistical analysis was carried out with Prism 8 software (GraphPad software) using one-way/two-way ANOVA followed by Bonferroni's post-test. For experimental designs with less than three groups and one variable we used the unpaired Student *T* test. In case of non-normal distribution, we applied the Kurskall–Wallis test followed by two-stage step-up method.

**Reporting summary**. Further information on research design is available in the Nature Research Reporting Summary linked to this article.

## Data availability

All antibodies and reagents used in the study are commercially available (see Supplementary Table 1). All the primer sequences are listed in Supplementary Information 1. All raw data generated or analyzed during this study are included in this published article (and its supplementary information files). Genomic data are available at GEO accession GSE157534, and GSE157536. Source data are provided for with this paper. All other data or resources are available from the corresponding author upon reasonable request. Source data are provided with this paper.

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

## Acknowledgements

We are thankful to D. Bonanomi, L. Muzio, and A. Menegon for sharing of reagents and all members of the Broccoli's lab for invaluable discussion. Human brain autoptic tissue samples were obtained by the Multiple Sclerosis and Parkinson's Tissue Bank, Imperial College London, UK. All the images were acquired at the Advanced Light and Electron Microscopy BioImaging Center (ALEMBIC), San Raffaele hospital, Milan. This work was supported by the Italian Ministry of Health (PE-2016-02363550) to V.B. and Fondazione Regionale per la Ricerca Biomedica, project Inflammapark (#1750818) to S.B.

## Author contributions

V.B and S.B. conceived this study. S.B., S.M. and V.B. designed the experiments. S.B. and S.M. with the help of S.G.G., G.O., A.I., M.N., E.M. and M.L. performed and analyzed all the experiments. L.M., E.B. and D.G. performed bioinformatics analysis. M.J.M. and F.N. supervised and analyzed scRNA sequencing. G.R., M.B., S.G. and C.M. analyzed scRNA sequencing datasets and contributed to immune cell characterization. V.B. and S.B. coordinated and supported the project and wrote the manuscript.

## Competing interests

The authors declare no competing interests.

## Ethic statement

The use of the human brain samples in this study was approved by the ethical committee of the Humanitas Clinical Center with the approval number 2561.
