## [Peer Review File · Nature Communications]

Reviewers' Comments:

Reviewer #1:

Remarks to the Author:

In this study, the authors have investigated the effect of selective expression of alpha-synuclein in microglial cells vs dopaminergic cells of mouse brain using a conditional cre-dependent lentiviral vector. They report progressive dopaminergic neurodegeneration also with microglia-specific overexpression. Extensive characterization by scRNA-sequencing reveals involvement of the innate and adaptive immune system. Pharmacological treatment with minocycline or an iNOS inhibitor reduce dopaminergic cell death in this model.

Although the study is novel, interesting, and extensive, there are several concerns and questions remaining.

Major remarks:

- Although the overexpression of a-synuclein specifically in microglial cells is interesting from a mechanistic point of view, it does not correlate with the situation in PD patients, where more likely a-synuclein can be released from dopaminergic neurons and then taken up by microglia. As far as we know, there is no evidence of selective aSYN accumulation in microglia in PD brain
- Cautiousness is needed in the author's interpretation on the role of synuclein oligomers. It has been shown that other types of synuclein assemblies, such as different types of fibrils, can act as activators of immune cells, including microglia.
- At the beginning of the study, the authors compare cre-inducible AAV and LV vectors and claim that only LV provide selective expression. However, this might be the result of the typical lower titers and therefore lower expression levels of the LV compared to AAV. No information is provided on the viral particle load of the AAV injected in this study compared to the LV. Reducing AAV titers might result in apparent similar specificity
- Throughout the manuscript the quantifications of co-localization or co-occurrence of fluorescent markers (or double positive cells) are often expressed as a percentage number of cells, instead of the Mander's coefficient. I assume the authors used Mander's overlap (MOC) as their method for quantification. Expressing MOC as percentage of positive cells is then not entirely correct. Based on the presented data, I suspect the actual number of double positive cells in fact to be higher (the mander's coefficient does not take into account what is a cell and what not). The text would need rephrasing to better reflect this.
- In the first part of the study, the authors compare the CX3CR1-creERT2 (CX3CR1-SNCA) and DAT-cre (DAT-SNCA) mice. However, a control viral vector encoding an unrelated protein such as dGFP is mentioned only once (p5), and no figures are provided for this control. In several figures of the study, 'control' means probably uninjected side, but this is not mentioned very clearly
- The authors demonstrate considerable DA cell loss after a-synuclein expression in microglia. To be sure that there is genuine DA cell loss instead of loss of TH expression, another independent DA or panneuronal marker should be used. Do the DA cells display any a-synuclein pathology in these conditions?
- The observation of increased CD22 expression in microglia in the experimental model is an interesting observation. The authors try to validate this finding by assessing the presence of CD22 in PD patient brain. There is positive detection of CD22 but it seems that the expression of CD22 is leukocytic, with a morphology that resembles more that of B cells (where CD22 is highly expressed) instead of microglia. Therefore, this does not convincingly corroborate the findings of elevated CD22 expression in microglia of the experimental model. The authors should show that the elevated CD22 expression is in microglia in PD patient brain, and not B-cells. This also should be adapted in the discussion.
- It is not so clear why in figure 6 the combination of a-synuclein with IFN γ is chosen and not another cytokine shown to be increased from the single-cell analysis

- In figure 6, the switch to treatment with fibrils instead of with LV is not very well motivated, since all in vivo experiments use LV-mediated expression of aSYN. Were the effects of alpha-synuclein expressing LVs compared with PFF stimulation of primary microglial cells in the absence of interferon? Figure 6A only shows the results of both conditions in the presence of interferon? In addition, the condition LV+ IFN γ is missing in B and C

Minor remarks

- Why have the authors chosen to use the A53T a-synuclein mutant and not the wild-type protein?
- P6 line 175 the authors mention, "... given the low immunogenicity achieved with the alphaSYN antibody ... ", but it is not mentioned why there is low immunogenicity?
- P7 'Increased levels of some of these genes in CX3CR1-SNCA relative to either CX3CR1-SNCA or control tissues was confirmed' should be 'relative to either DAT-SNCA'
- Figure 1 : in DAT-SNCA animals, the authors mention the aSYN expression is limited to the TH+ neurons, but it seems that there is more blue signal than red. Also the picture of the CX3CR1-SNCA condition does not show nice DA cell bodies. In this condition, there are also blue cells that do not colocalize with IBA1.
- The text should be revised by a native English speaker

Reviewer #2:

Remarks to the Author:

The main finding of this study, i.e. loss of nigral dopaminergic neurons as a consequence of microglial accumulation of alpha-synuclein (As), is interesting and novel. Characterization of the effects of As accumulation on microglial morphology/function and tissue immune response is also thorough and potentially relevant to pathogenetic processes in humans. Important concerns are raised, however, by the design of this study and interpretation of some of its results. In particular:

1. The rationale for using lentiviruses for targeted transgene expression is well documented. On the other hand, it is not clear why Authors decided to overexpress human mutated rather than wild-type As. It is difficult to extrapolate data obtained with this rare form of As in terms of their relevance to idiopathic Parkinson's disease (PD).
2. Control experiments are carried out using cre-inducible lentiviruses expressing destabilized GFP. Authors mention (but do not show) that these control injections resulted in transgene expression of similar levels compared to injections of lentiviruses carrying mutated As. Transgene expression levels should be carefully documented and reported in the Results section. One of the questions raised by this work concerns the specificity of the effects of As accumulation within microglia. In other words, are the effects shown in this manuscript specifically related to As accumulation, or does accumulation of other proteins within microglia also results in similar changes? Destabilized GFP is characterized by rapid turnover that would prevent accumulation of the transgene. Is this the most appropriate control to account for the non-specific effects of microglial transgene accumulation?
3. One of the interesting findings of this study concerns the possibility that accumulation of As within microglia is itself capable of activating these cells; this activation may in turn result in a "downstream" neurodegenerative effect on nigral dopaminergic neurons. Changes in microglial morphology and markers of microglial activation as well as assessment of tissue immune response were studied at 5 weeks after viral administration. At this time point, 30% of dopaminergic cells are already dead, and this death would likely have contributed to the activation of microglia. If the purpose of these experiments is to characterize the changes in microglial morphology and function that are directly related to As accumulation (and then responsible for neuronal death), analyses should also be made at an earlier time point, prior to or at the beginning of cell death.

4. The conclusion “microglial-specific As accumulation triggers a selective loss of nigral DA neurons...”, which is apparently supported by data showing no cell death in the substantia nigra pars reticulata, is not correct since a significant neuronal loss seems to occur also in the VTA (figure 3). Stereological quantification of DA neurons in the VTA should be provided.

5. Quantification of DA neurons is shown based on their TH phenotype. Authors should rule out the possibility that lower cell counts may be due (at least in part) to TH downregulation rather than actual neurodegeneration.

6. The statement “DA neurons did not show any evidence of As accumulation” in the Discussion is not supported by actual data in the Results section. Did these neurons contain mutated human As or show evidence of As pathology? Were neuronal levels of endogenous As similar to control values? Also, did peripheral immune cells present in the nigral tissue accumulate As and/or phosphorylated As?

7. The concept that a “primary” neuroinflammatory process involving sustained microglial activation leads to death of nigral dopaminergic neurons is not new. It has been extensively documented, for example, in animal models using systemic LPS injections. No reference to this earlier work is made in the manuscript, however. Do Authors think that the mechanisms involved in microglial-mediated neurodegeneration in their As model are different than the mechanisms described in earlier models? Studies on microglial morphology and function after As accumulation may be significantly strengthened if they included a comparison with microglial changes under other paradigms of direct microglial activation and microglial-induced neurodegeneration. This comparison would help address the key issue of “specificity” (or lack of it) of As-related microglial and neuronal toxicity.

8. In vitro evidence shown in this manuscript suggests that the pathological/toxic effects of exposure to As fibrils were the same as the effects caused by mere As accumulation. Were levels of unmodified and phosphorylated As similar under these two experimental paradigms? In the Introduction and Discussion of the paper, Authors underscore that As oligomers play a critical role in the pathological processes triggered by exposure to As fibrils. What evidence supports a role of As oligomers in this study?

9. The pathophysiological relevance that the work presented in this manuscript may have to PD should be discussed with greater caution. What evidence in PD supports a considerable load of As within microglia as a primary pathogenetic event? How could the results of this study, which were obtained using a mutated form of As, be extrapolated to PD pathogenetic processes? Sentences such as “abnormal CD22 increase is a pathological feature conserved between CXCR1-SNCA mice and the human disease” seem overstatements.

Reviewer #3:

Remarks to the Author:

In their manuscript entitled «Microglia-specific overexpression of alpha-Synuclein leads to severe dopaminergic neurodegeneration by phagocytic exhaustion and oxidative toxicity”, the authors used a viral approach to drive the expression of the human SNCA mutant gene in either dopaminergic (DA) neurons or microglia from the substantia nigra. They then used a large range of approaches to assess the in-vivo consequences of these cell-specific expression on different features that are relevant to Parkinson disease (PD). More specifically, they focused on neuroinflammation related processes and their effects on DA neurons survival. They identified molecular and cellular pathways that are likely playing significant roles in DA neurodegeneration in PD. Finally, they made the proof-of-principle that targeting these pathways represent interesting therapeutic perspectives for the treatment of this disease.

Overall, this is a well performed study with clear objectives and results. However, some aspects of the study, in particular scRNA-seq results, would merit further analyses. Importantly, as their main results relate to the expression of the human SNCA mutant gene in microglia the authors should

discuss its clinical relevance for Parkinson disease.

Specific comments:

1.Line 169: The authors state that their viral approach did not lead to overaccumulation of the mutated α SYN in the transduced cells, whether these latter are neurons or microglia. They indeed should show that α SYN is expressed at about endogenous levels. Additionally, they should indicate, at least 5 weeks after surgery, to which extend lentiviral transduction in the substantia nigra affects inflammation and DA neurons survival.

2.Line 207: The authors should explain why DA neurons are counted in the whole substantia nigra while NeuN+ neurons quantification is restricted in SN pars reticulata.

3.Line 210: The authors state that "DA neuronal degeneration was comparable both in numbers and temporal dynamics between treated DAT-SNCA and CX3CR1-SNCA mice", however Fig. 3I shows a much stronger effect for DA survival in DA-SNCA compared to CX3CR1-SNCA mice. It would also have been interesting to test, 5 weeks after viral transduction, whether expression of mutated α SYN in both DA neurons and microglia has a synergetic effect on DA neurons death.

4.Line 232: The authors state that "...only in CX3CR1-SNCA mice the microglia exhibited a strong inflammatory state". This clearly not true. Indeed, hierarchical clustering in supp-Fig 5C and supp-Fig 5D clearly show intermediate inflammatory states in DAT-SNCA mice. Similarly, T-cells infiltration is higher in DAT-SNCA compared to control mice (supp-Fig 6).

5.Lines 250-316: It is not clear why the authors did not analyze all the cells from their scRNA-seq experiment together in the same UMAP plot? Presenting the different cell populations on the same plot would make it possible to highlight the cell populations and sub-populations that are common to the two experimental conditions and those that are different.

This part of the results is also less clearly presented, the genes mentioned in the text are not always those shown on the figures. As a typical example of this, lines 262 the authors state that they identified Cluster 1-3 as microglial cells because they expressed the cardinal genes *Olfml3*, *Gpr34*, *Siglech* and *Cx3cr1*; and referred to Fig. 4B,C. However, of those only *Cx3cr1* is shown in Fig. 4C. Similarly, Cluster 4 is not clearly defined and with discrepancies between the figures and the text.

Overall, this part should be revised to ensure better consistency between the text and the figures. In the figures, identification of the cell populations/subpopulation is not clear, and there are clearly offsets between the colors indicating the different cell populations and the actual names of the populations/subpopulations.

The authors should also compare and comment how the microglial signature they unravel overlaps with other microglia signatures in disease states (i.e. comparison with DAMs, MGnD, ARMs...).

6.Figure 6: From the text, it seems that effects of LV transduction and α SYN fibrils application was compared, however the second line of the heatmap indicates treatment with IFN γ . The legend of the figure does not shed light the actual conditions compared.

7.Line 528: The authors referred to B6;FVB-Tg(*Aldh1l1-cre/ERT2*)1Khakh/J as being the CX3CR1-creERT2 mice.... Hopefully this is a copy/paste mistake.... Additionally, in this paragraph of the materials and methods section, the authors should provide the legal agreement number of their animal facility as well as the legal agreement number for their specific project.

8.Line 673: It is not clear why the authors did not use the same type of correction for multiple testing (i.e. they used either Bonferroni or two-stage linear step-up procedure of Benjamini, Krieger and Yekutieli) in the post-hoc tests they performed in ANOVA analyses.

Minor comments:

1.The authors should comply the official nomenclature when naming mouse & human genes and proteins (DOI: 10.4161/auto.20665)

2.Line 180: The authors should refer to Fig. 2, not Fig 1.

3.Supp figure 5: For readability of the figures, it would be better to keep the same color codes for the experimental groups throughout the manuscript.

4.Line 288: "Chatepsin" should be replaced by "cathepsin"

5.Line 446: the reference is truncated

6.Lines 572-576: This part of the text appears in red...

7.Line 675: A space is missing between "using" and "one"

8.Line 724: "CXCR1" should be replaced by "CX3CR1"

9.Line 727: "n=8" should be replaced by "n=6-8"

10.Line 741: spelling error in "transduction"

11. Line 788: please include all the p-values shown on the graph
12. Line 857: Panel Supp-Fig5E is not described in the legend of the figure. A diverging scale bar will help appreciate the difference of expression between the different experimental conditions
13. Line 873 & 879: Supp-Fig7 and 8: Insert scale bars on the images
14. Line 886: include the statistics of the graph shown in supp-Fig9

REVIEWER COMMENTS

Reviewer #1 (Remarks to the Author):

In this study, the authors have investigated the effect of selective expression of alpha-synuclein in microglial cells vs dopaminergic cells of mouse brain using a conditional cre-dependent lentiviral vector. They report progressive dopaminergic neurodegeneration also with microglia-specific overexpression. Extensive characterization by scRNA-sequencing reveals involvement of the innate and adaptive immune system. Pharmacological treatment with minocycline or an iNOS inhibitor reduce dopaminergic cell death in this model. Although the study is novel, interesting, and extensive, there are several concerns and questions remaining.

Major remarks:

- Although the overexpression of a-synuclein specifically in microglial cells is interesting from a mechanistic point of view, it does not correlate with the situation in PD patients, where more likely a-synuclein can be released from dopaminergic neurons and then taken up by microglia. As far as we know, there is no evidence of selective aSYN accumulation in microglia in PD brain.

The reviewer raises an important issue that is crucial for all the experiments regarding PD. The attempt to get as close as possible to the pathophysiological process in human patients face numerous challenges due to the species-specific differences in brain complexity and time course of degeneration. This particular aspect is well known to the authors, in fact, the objective of this study is limited to dissect the role of α SYN-dependent microglia inflammation in DA neuronal degeneration. We are aware that PD is caused by multiple noxious processes that are entangled together, but we found the need to accurately analyze the specific contribution of microglia activation to the disease onset and progression. This is due, in part, to the lack in literature of exhaustive descriptions of the toxicity exerted by microglia and its specific impact on neurodegeneration *in vivo*. To our knowledge, all the available models are not able to distinguish between the toxicity acting directly on the neurons caused by external stimuli (6-OHDA, MPTP, AAV-mediated overexpression of α SYN, PPF injection) and the detrimental effect promoted by inflamed microglia. Recently, Olanow and co-workers (Brain 2019) have demonstrated the presence of diffuse microglia activation in the brain of PD patients grafted with healthy neurons lasting for all the observation period (from 18 months up to 16 years). Interestingly, they described microglia activation long before the appearance of α SYN accumulation pathology, suggesting an integral role of microglia in an early phase of the disease. From their analysis it appears that microglia is the first cell type that accumulates α SYN. In overall, there is an increasing number of studies showing that many PD-related genes such as PINK1, PRKN, GBA and LRRK2 have an autophagy-related role in microglial cells (i.e. Gan-Or et al., Autophagy 2015). For instance, mutant PINK1 impacts on microglia behavior causing inflammation-mediated-neuronal death (Sun et al., Sci Rep 2018). Far from proposing a new PD mouse model, we believe that this setting will enable to assess the specific contribution of microglial dysregulation on the survival and function of DA neurons avoiding confounding effects originated by concurrent pathological mechanisms occurring in other cell types.

- Cautiousness is needed in the author's interpretation on the role of synuclein oligomers. It has been shown that other types of synuclein assemblies, such as different types of fibrils, can act as activators of immune cells, including microglia.

We fully agree with the reviewer's comment. We revised the manuscript avoiding to correlate α SYN pathology to its oligomeric state, but more widely to its aggregated species.

- At the beginning of the study, the authors compare cre-inducible AAV and LV vectors and claim that only LV provide selection expression. However, this might be the result of the typical lower titers and therefore lower expression levels of the LV compared to AAV. No information is provided on the viral particle load of the AAV injected in this study compared to the LV. Reducing AAV titers might result in apparent similar specificity

We thank the reviewer for this important suggestion. The results presented in the original work were obtained by injecting 2 μ l of AAVs or LVs with titers 10e10 vg/ml or 10e8 pfu/ml, respectively. To further investigate this issue, we performed additional injections in DAT-cre mice using the same AAV-FLEX-GFP virus with the lower titers 10e9 and 10e8 vg/ml, with the last dosage comparable with that used for LVs. After immunofluorescent staining, although TH+ neurons were strongly positive, the signal was also noticeable in the surrounding cells indicating a lack of selectivity of GFP expression in the target tissue (new Suppl. Fig. 1). Awkwardly, the direct GFP signal without antibody amplification appeared mostly confined to TH+ neurons and this can mislead the interpretation of the results (data not shown). Given that most of our work relies on α SYN overexpression, even its smallest leakage in unwanted cells would compromise the accuracy of the outcome and the value of this study. For this reason, we consider the use of LVs a much safer and more reliable strategy respect to AAVs.

- Throughout the manuscript the quantifications of co-localization or co-occurrence of fluorescent markers (or double positive cells) are often expressed as a percentage number of cells, instead of the Mander's coefficient. I assume the authors used Mander's overlap (MOC) as their method for quantification. Expressing MOC as percentage of positive cells is then not entirely correct. Based on the presented data, I suspect the actual number of double positive cells in fact to be higher (the mander's coefficient does not take into account what is a cell and what not). The text would need rephrasing to better reflect this.

The co-localization of markers is calculated throughout the manuscript by using the Manders' Coefficient using fixed thresholds for all the images. The outcomes, calculated with JACoP plugin for ImageJ, represent the fraction of image A overlapping image B and not, as the reviewer notices the percentage of co-localized cells. We better described this in the revised manuscript.

- In the first part of the study, the authors compare the CX3CR1-creERT2 (CX3CR1-SNCA) and DAT-cre (DAT-SNCA) mice. However, a control viral vector encoding an unrelated protein such as dGFP is mentioned only once (p5), and no figures are provided for this control. In several figures of the study, 'control' means probably uninjected side, but this is not mentioned very clearly

We apologize if the choice of the controls was not clearly detailed. In all our experiments the "control" condition refers to the CX3CR1-creERT2 mice injected with the cre-inducible destabilized GFP (dGFP). We selected this condition to better recapitulate the injection, infection and protein expression of a transgene in the microglia. Noticeably the accumulation of GFP, but not of dGFP, does elicit a significant toxicity in dopaminergic neurons (Suppl. Fig. 3a). For this reason, we opted to express the dGFP in microglia. Nonetheless, in subsequent studies performed to answer to Reviewer 2, we also found that microglia expressing GFP remain in a resting state without developing any sign of inflammation (see Fig. below for reviewers). Thus, only overexpression of SNCA, but not dGFP or GFP, induces a change in microglia homeostasis leading to a severe neuroinflammation state.

- The authors demonstrate considerable DA cell loss after a-synuclein expression in microglia. To be

sure that there is genuine DA cell loss instead of loss of TH expression, another independent DA or panneuronal marker should be used. Do the DA cells display any a-synuclein pathology in these conditions?

We took this opportunity offered by the reviewer's comment to better describe the neurodegeneration in our models through an accurate quantification of NeuN positive neurons in both the SNc and VTA (also in response to Reviewer 2). Revised Suppl. Fig. 7 shows the stereological counting of NeuN positive nuclei in the SNc. The loss of NeuN staining follows the same time course of the previous TH quantification.

- The observation of increased CD22 expression in microglia in the experimental model is an interesting observation. The authors try to validate this finding by assessing the presence of CD22 in PD patient brain. There is positive detection of CD22 but it seems that the expression of CD22 is leukocytic, with a morphology that resembles more that of B cells (where CD22 is highly expressed) instead of microglia. Therefore, this does not convincingly corroborate the findings of elevated CD22 expression in microglia of the experimental model. The authors should show that the elevated CD22 expression is in microglia in PD patient brain, and not B-cells. This also should be adapted in the discussion.

We generated additional images that better corroborate the presence of CD22+ microglia in the autaptic brain tissue of PD patients. These results were obtained by performing double immunofluorescence for IBA1 and CD22 in both PD and healthy control samples and imaging at high magnification view (63x). We added these results in the revised Suppl. Fig. 13.

- It is not so clear why in figure 6 the combination of a-synuclein with IFN γ is chosen and not another cytokine shown to be increased from the single-cell analysis

We apologize if this point was not clear enough. As we better explained in the revised manuscript, we decided to add IFN γ since it is one of the cytokines mostly expressed by T and NK cells as found in the single cell transcriptomic analysis. Moreover, IFN γ is well known for its ability to stimulate the activation, proliferation and NO production of microglia (Thuy-Truc Ta et al., 2019, PNAS).

- In figure 6, the switch to treatment with fibrils instead of with LV is not very well motivated, since all in vivo experiments use LV-mediated expression of aSYN. Were the effects of alpha-synuclein expressing LVs compared with PFF stimulation of primary microglial cells in the absence of interferon? Figure 6A only shows the results of both conditions in the presence of interferon? In addition, the condition LV+ IFN γ is missing in B and C.

To address this concern, we performed more in vitro experiments by infecting the primary microglia with the LV expressing the mutant α Syn. The results obtained with this model provides the evidence that either the α Syn in fibrils or expressed by LVs lead to impaired phagocytosis at a comparable degree (revised Fig. 6).

Minor remarks

- Why have the authors chosen to use the A53T a-synuclein mutant and not the wild-type protein?

We decided to use the human A53T mutated α Syn instead of the wild type form for its superior capacity to accumulate in the targeted cells. It is well documented that the A53T mutated α Syn is

responsible for the impairment of the cellular clearance mechanisms, in particular the chaperon mediated autophagy (Cuervo et al., Science 2004) and it induces alterations of the ubiquitin-dependent degradation system (Stefanis et al., J Neurosci. 2001; recently in McKinnon et al., Acta Neuropat. Comm. 2020). These features can boost the accumulation of α Syn, thus promoting the formation of aggregates in a time course and reproducibility that better meets the experimental needs. Anyway, the pathological mechanisms are not expected to be different between the two forms, but simply the A53T mutated variant accelerates the overall pathological process. For this reason, the overexpression of A53T aSyn is commonly used to test pathological mechanisms or validate therapeutic treatments. A small list of studies based on the same vector are as follows: Bu et al., Neurotherapeutics 2021; Beal et al. Acta Neuropathol. Commun. 2021; Schneider et al., Sci Rep 2019; Arawaka et al., Acta Neuropathol. Commun. 2017; Gleave et al., Neurobiol Dis 2017; Chansel-Debordeaux et al., Gene Ther. 2017; Bourdenx et al., Acta Neuropathol. Commun. 2015; He et al., Mol Neurobiol 2016; Gaugler et al., Acta Neuropathol 2012; Chung et al., J. Neurosci. 2009. Moreover, among the most widely used mouse models for studies in PD are the A53T aSyn overexpressing transgenic mice (i.e. line M83 by the V. Lee's lab, JAX #004479).

Nonetheless, prompted by the comment of the reviewer, we generated a FLEX LV for the conditional expression of the wild type aSyn and injected in DAT-cre and CX3CR1-creERT2 mice. Interestingly, selective expression of the wild type aSyn in microglia for 5 weeks was sufficient to induce a significant loss of nigral DA neurons. Thus, either the wild type or the mutated form of aSyn when expressed in microglial cells are causing neurodegeneration, although the mutant variant has a stronger effect accelerating the neuronal loss at least at 5 weeks. We added these results in the revised manuscript illustrated in Suppl. Fig. 9.

- P6 line 175 the authors mention, "... given the low immunogenicity achieved with the alphaSYN antibody ... ", but it is not mentioned why there is low immunogenicity?

Indeed, we realized that this sentence was not clear and could be misinterpreted. We wanted to refer to the fact that endogenous α Syn is barely detectable with this antibody, even in cells that express Syn at reasonable levels like in human neuronal progenitors. Conversely, it delivers a good signal only when α Syn levels are extremely high as after gene overexpression. We amended the sentence for better clarity.

- P7 'Increased levels of some of these genes in CX3CR1-SNCA relative to either CX3CR1-SNCA or control tissues was confirmed' should be 'relative to either DAT-SNCA'

We corrected the typo.

- Figure 1: in DAT-SNCA animals, the authors mention the aSYN expression is limited to the TH+ neurons, but it seems that there is more blue signal than red. Also, the picture of the CX3CR1-SNCA condition does not show nice DA cell bodies. In this condition, there are also blue cells that do not colocalize with IBA1.

This unbalance is due to a slight difference in the levels of the two proteins along the neurites. In fact, TH signal is maximal in the cell body and lower in the projections, on the contrary α Syn accumulates at high concentrations in both the soma and neurites. We provided new images included in the revised Suppl. Fig. 2a-h where the immunostaining signals are particularly well delineated.

- The text should be revised by a native English speaker

The revised text has been edited by a professional service.

Reviewer #2 (Remarks to the Author):

The main finding of this study, i.e. loss of nigral dopaminergic neurons as a consequence of microglial accumulation of alpha-synuclein (As), is interesting and novel. Characterization of the effects of As accumulation on microglial morphology/function and tissue immune response is also thorough and potentially relevant to pathogenetic processes in humans. Important concerns are raised, however, by the design of this study and interpretation of some of its results. In particular:

1. The rationale for using lentiviruses for targeted transgene expression is well documented. On the other hand, it is not clear why Authors decided to overexpress human mutated rather than wild-type As. It is difficult to extrapolate data obtained with this rare form of As in terms of their relevance to idiopathic Parkinson's disease (PD).

We decided to use the human A53T mutated α Syn instead of the wild type form for its superior capacity to accumulate in the targeted cells. It is well documented that the A53T mutated α Syn is responsible for the impairment of the cellular clearance mechanisms, in particular the chaperon mediated autophagy (Cuervo et al., Science 2004) and it induces alterations of the ubiquitin-dependent degradation system (Stefanis et al., J Neurosci. 2001; recently in McKinnon et al., Acta Neuropat. Comm. 2020). These features can boost the accumulation of α Syn, thus promoting the formation of aggregates in a time course and reproducibility that better meets the experimental needs. Anyway, the pathological mechanisms are not expected to be different between the two forms, but simply the A53T mutated variant accelerates the overall pathological process. For this reason, the overexpression of A53T aSyn is commonly used to test pathological mechanisms or validate therapeutic treatments. A small list of studies based on the same vector are as follows: Bu et al., Neurotherapeutics 2021; Beal et al. Acta Neuropathol. Commun. 2021; Schneider et al., Sci Rep 2019; Arawaka et al., Acta Neuropathol. Commun. 2017; Gleave et al., Neurobiol Dis 2017; Chansel-Debordeaux et al., Gene Ther. 2017; Bourdenx et al., Acta Neuropathol. Commun. 2015; He et al., Mol Neurobiol 2016; Gaugler et al., Acta Neuropathol 2012; Chung et al., J. Neurosci. 2009. Moreover, among the most widely used mouse models for studies in PD are the A53T aSyn overexpressing transgenic mice (i.e. line M83 by the V. Lee's lab, JAX #004479).

Nonetheless, prompted by the comment of the reviewer, we generated a FLEX LV for the conditional expression of the wild type aSyn and injected in DAT-cre and CX3CR1-creERT2 mice. Interestingly, selective expression of the wild type aSyn in microglia for 5 weeks was sufficient to induce a significant loss of nigral DA neurons. Thus, either the wild type or the mutated form of aSyn when expressed in microglial cells are causing neurodegeneration, although the mutant variant has a stronger effect accelerating the neuronal loss at least at 5 weeks. We added these results in the revised manuscript illustrated in Suppl. Fig. 9.

2. Control experiments are carried out using cre-inducible lentiviruses expressing destabilized GFP. Authors mention (but do not show) that these control injections resulted in transgene expression of similar levels compared to injections of lentiviruses carrying mutated As. Transgene expression levels should be carefully documented and reported in the Results section. One of the questions raised by this work concerns the specificity of the effects of As accumulation within microglia. In other words, are the effects shown in this manuscript specifically related to As accumulation, or does accumulation of other proteins within microglia also results in similar changes?

We performed new qPCRs on infected nigral tissue lysates to amplify the viral WPRE sequence which is in common between the AAV-dGFP and AAV-SNCA vectors showing that their viral transgene expression is comparable between the two conditions (Suppl. Fig. 2a). While we opted for dGFP for its lack of toxicity in DA neurons (Suppl. Fig. 2b), we performed new experiments expressing GFP exclusively in the microglia in CX3CR1-cre mice. As shown in the revised Suppl. Fig. 6, GFP is unable to trigger an inflammatory response in microglia 5 weeks after viral delivery. Thus, only overexpression of SNCA, but not dGFP or GFP, induces a change in microglia homeostasis leading to a severe neuroinflammation state.

- Destabilized GFP is characterized by rapid turnover that would prevent accumulation of the transgene. Is this the most appropriate control to account for the non-specific effects of microglial transgene accumulation?

We opted for the use of the destabilized GFP (dGFP) due to the documented immunogenicity and cytotoxicity related to the expression of GFP (for a review Ansari et al., 2016). In particular, in our hands, the infection of the DA neurons with the LV-GFP-FLEX in DAT-cre mice is sufficient to induce a significant neurodegeneration (Suppl. Fig. 2b). We don't know at this point why the GFP overexpression is toxic to DA neurons but not to other neuronal cell types, but for this reason we considered the GFP to be inappropriate as a control. However, intrigued by the reviewer's concern we tested the eventual effects of expressing GFP in microglia. As shown in the revised Suppl. Fig. 6, selective overexpression of GFP in microglia achieved by injecting the LV-GFP-FLEX in CX3CR1-cre mice did not trigger any evident microglial activation. These results indicate that, differently from what observed in DA neurons, microglia homeostasis is not altered by GFP overexpression at least after 5 weeks.

3. One of the interesting findings of this study concerns the possibility that accumulation of As within microglia is itself capable of activating these cells; this activation may in turn result in a "downstream" neurodegenerative effect on nigral dopaminergic neurons. Changes in microglial morphology and markers of microglial activation as well as assessment of tissue immune response were studied at 5 weeks after viral administration. At this time point, 30% of dopaminergic cells are already dead, and this death would likely have contributed to the activation of microglia. If the purpose of these experiments is to characterize the changes in microglial morphology and function that are directly related to As accumulation (and then responsible for neuronal death), analyses should also be made at an earlier time point, prior to or at the beginning of cell death.

This issue raised by the reviewer is of particular relevance. For this reason, we analyzed new animals at 2 weeks after LV infections. At this early time point evident microgliosis was already observed which preceded the advent of neurodegeneration (Suppl. Fig. 5a,c). We also performed additional analyses on the neuroinflammation state 15 weeks after viral transduction. Notably, enhanced inflammation was still detectable at this late period (Suppl. Fig. 5b,c), suggesting that α Syn accumulation fuels a long-term inflammatory condition. A major reason why we decided to carry out our analysis at 5 weeks was to be sure to avoid any confounding contribution of the surgical procedure by itself on the tissue inflammation as it occurs transiently soon after the manipulation.

4. The conclusion "microglial-specific As accumulation triggers a selective loss of nigral DA neurons...", which is apparently supported by data showing no cell death in the substantia nigra pars reticulata, is not correct since a significant neuronal loss seems to occur also in the VTA (figure 3). Stereological quantification of DA neurons in the VTA should be provided.

We quantified by stereological counting the number of TH positive neurons in VTA. As noticed by the reviewer we have indeed neurodegeneration also in the VTA that follows a time course comparable to the cell loss in the SN. We added these results in Fig. 3 and amended the description to better detail these findings.

5. Quantification of DA neurons is shown based on their TH phenotype. Authors should rule out the possibility that lower cell counts may be due (at least in part) to TH downregulation rather than actual neurodegeneration.

To investigate this important aspect, we performed quantifications of NeuN positive neurons in the SNc. These measurements confirmed the progressive neuronal loss in this area after the treatments (Suppl. Fig. 7).

6. The statement “DA neurons did not show any evidence of As accumulation” in the Discussion is not supported by actual data in the Results section. Did these neurons contain mutated human As or show evidence of As pathology? Were neuronal levels of endogenous As similar to control values? Also, did peripheral immune cells present in the nigral tissue accumulate As and/or phosphorylated As?

To further support our conclusions, we provided new high-magnification (63X) images where human α Syn is undetectable in TH positive neurons while it accumulates in Iba1 positive microglia in CX3CR1-SNCA animals (Suppl. Fig. 2a-h). On the other hand, α Syn accumulation in DA neurons is readily detectable in DAT-SNCA mice with an identical detection procedure (Suppl. Fig. 2a-h). As suggested by the reviewer, we also confirmed that mouse α Syn endogenous levels are not changed between the different treatments (Figure below for the reviewer). We also performed new staining for human α Syn in CD3+ T cells, failing to detect any sign of exogenous α Syn accumulation (Suppl. Fig. 11e).

Figure for reviewers: Mouse endogenous α Syn levels are not changed between the different treatments. Immunostaining for TH and the endogenous mouse α Syn (D37A6 antibody, Cell Signaling) in nigral tissue of mice 5 weeks after treatments. Endogenous α Syn protein levels are comparable in TH+ neuronal somata of the different experimental mouse models.

7. The concept that a “primary” neuroinflammatory process involving sustained microglial activation leads to death of nigral dopaminergic neurons is not new. It has been extensively documented, for example, in animal models using systemic LPS injections. No reference to this earlier work is made in the manuscript, however. Do Authors think that the mechanisms involved in microglial-mediated neurodegeneration in their As model are different than the mechanisms described in earlier models? Studies on microglial morphology and function after As accumulation may be significantly strengthened if they included a comparison with microglial changes under other paradigms of direct microglial activation and microglial-induced neurodegeneration. This comparison would help address the key issue of “specificity” (or lack of it) of As-related microglial and neuronal toxicity.

We apologize for the lack of references on the previous literature on LPS treatments. The LPS models are of great interest for understanding the role of systemic inflammation in the neurodegenerative disorders. The correlation between inflammation and some loss of DA neurons has been previously described. The pioneer studies performed in rats with nigral LPS described the microglia activation and the dopaminergic cell loss, which are accompanied by the increased IL1b, TNFa and IL6 among other factors (Castano et al., J. Neurosci 1998; Lu et al., Neuroscience 2000; Hernandez-Romero et al., J Neurochem 2008; Arimoto et al., Neurob Aging 2007). In the attempt to study the impact of the infectious disease in neurodegenerative disorders, LPS has been also administered intraperitoneally or intravenously. By taking this route of administration, LPS is able to upregulate the CD14 present not only in microglia but also in monocytes and macrophages (Lacroix et al., 1998, Brain Pathology). In this view, one can compare the mechanisms underlying the LPS-mediated neurodegeneration with microglial α Syn accumulation and conclude that both converge on the same pathological process (microglia activation). However, we have to keep in mind that systemic LPS acts in several brain region and, for this reason it is also widely used for models of amyloidosis and inflammation in Alzheimer’s disease (Miklossy, J Alzheimer Dis 2008). Moreover, due to intraperitoneal administration, LPS can induce changes in peripheral blood mononuclear cells, a phenomenon that have profound impact on blood-brain barrier permeability and on neurovascular network (Banks et al., J Neuroinflam 2015). This puzzling picture makes difficult to isolate the microglia contribution from the pleiotropic effects exerted by LPS. In contrast, in our approach we genetically tuned α Syn expression only in the brain microglia avoiding any confounding effects from other brain cell types or peripheral triggers. This model features the early events of α Syn engulfment that recapitulate the temporal evolution observed in patients (i.e. Olanow et al., Brain 2019). We agree that a direct comparison of our model with the current LPS protocol would be interesting, but it is beyond the aim of this study focused on understanding the pathological events triggered by the exclusive α Syn accumulation in microglia and leading to severe DA neuronal loss.

8. In vitro evidence shown in this manuscript suggests that the pathological/toxic effects of exposure to As fibrils were the same as the effects caused by mere As accumulation. Were levels of unmodified and phosphorylated As similar under these two experimental paradigms? In the Introduction and Discussion of the paper, Authors underscore that As oligomers play a critical role

in the pathological processes triggered by exposure to As fibrils. What evidence supports a role of As oligomers in this study?

Prompted by this comment of the review, we analyzed phosphorylated α Syn in microglia exposed to LV-SNCA showing the presence of pS129 α Syn dots in the majority of the infected microglial cells as occurring using α Syn fibrils. The results are presented in the figure below for the reviewers. Our study is not conceived to determine the specific role of the different α Syn assemblies in this pathological process. For this reason, we modified the sentences highlighted by the review switching to the more general terminology of α Syn aggregated species.

Figure for reviewers: *aSYN/pS129syn double immunostaining on microglia transduced with the α Syn expressing LV. Cell expressing the viral α Syn transgene show the development of pS129Syn puncta indicating the formation of α Syn aggregates.*

9. The pathophysiological relevance that the work presented in this manuscript may have to PD should be discussed with greater caution. What evidence in PD supports a considerable load of As within microglia as a primary pathogenetic event? How could the results of this study, which were obtained using a mutated form of As, be extrapolated to PD pathogenetic processes? Sentences such as “abnormal CD22 increase is a pathological feature conserved between CXCR1-SNCA mice and the human disease” seem overstatements.

In line with this comment of the reviewer, we made clear that we did not intend to generate another generic model of the disease, but instead to dissect the relevance of α Syn accumulation in microglia devoid of any other confounding pathological process in vivo. Increasing evidence are pointing to an early involvement of neuroinflammation during the pathological progression in PD. As for instance, a recent paper by Olanow et al. (Brain, 2019) described how inflammation took place long before the accumulation of α Syn in grafted neurons in the striatum of PD patients. This description suggest that microglia can represent the first cell type sensing the presence of extracellular α Syn toxic species. On this same line, Haenseler and colleagues (Sci. Rep., 2017), by generating iPSC-derived macrophages from PD patients with SNCA gene triplication, observed increased intracellular α Syn protein levels and high inflammation state which correlated with an impaired phagocytic activity. A more recent study further extended these findings in vivo by showing how α Syn released by neurons is engulfed by microglia impairing autophagy and inducing a reactive state (Yue et al., Nat. Comm. 2020). We are aware that the load of α Syn triggered in our model is certainly higher to what is occurring in PD patients, but as it happens for the entirety of the PD mouse models, expression of the toxic agent is higher in order to overcome the insurmountable difference among the species in terms of disease time-course. This is the same reason that convinced us to use the

A53T mutated form of α Syn, chosen due to its ability to accelerate the accumulation of toxic species and their detrimental effects of the cellular processes.

Reviewer #3 (Remarks to the Author):

In their manuscript entitled «Microglia-specific overexpression of alpha-Synuclein leads to severe dopaminergic neurodegeneration by phagocytic exhaustion and oxidative toxicity”, the authors used a viral approach to drive the expression of the human SNCA mutant gene in either dopaminergic (DA) neurons or microglia from the substantia nigra. They then used a large range of approaches to assess the in-vivo consequences of these cell-specific expression on different features that are relevant to Parkinson disease (PD). More specifically, they focused on neuroinflammation related processes and their effects on DA neurons survival. They identified molecular and cellular pathways that are likely playing significant roles in DA neurodegeneration in PD. Finally, they made the proof-of-principle that targeting these pathways represent interesting therapeutic perspectives for the treatment of this disease.

Overall, this is a well performed study with clear objectives and results. However, some aspects of the study, in particular scRNA-seq results, would merit further analyses. Importantly, as their main results relate to the expression of the human SNCA mutant gene in microglia the authors should discuss its clinical relevance for Parkinson disease.

We thank the reviewer for appreciating our work.

Specific comments:

1.Line 169: The authors state that their viral approach did not lead to overaccumulation of the mutated α SYN in the transduced cells, whether these latter are neurons or microglia. They indeed should show that α SYN is expressed at about endogenous levels. Additionally, they should indicate, at least 5 weeks after surgery, to which extend lentiviral transduction in the substantia nigra affects inflammation and DA neurons survival.

To better investigate this issue, we performed qPCRs for either transgene (dGFP and SNCA) from nigral tissue lysates 2 weeks after viral delivery. This analysis confirmed that both transgenes are expressed to similar levels (Fig. Suppl. 2b). Figures 2 and 3 show that after lentiviral transduction with dGFP (control condition) no sign of inflammation or DA neuronal loss are detectable. In fact, in the dGFP (control) condition microglial morphology reflects a classical homeostatic state with no activation of inflammatory markers such as CD68 (Fig. 2). Moreover, numbers of DA neurons quantified by stereological counting are comparable with the uninjected control (data not shown).

2.Line 207: The authors should explain why DA neurons are counted in the whole substantia nigra while NeuN+ neurons quantification is restricted in SN pars reticulata.

Initially, this analysis was limited to ensure that neuronal loss was not detectable in the SN *pars reticulata*. We extended this characterization in the SN *pars compacta* showing a progressive loss of NeuN positive neurons and confirming previous results. These new results are presented in Suppl. Fig. 7a.

3.Line 210: The authors state that “DA neuronal degeneration was comparable both in numbers and temporal dynamics between treated DAT-SNCA and CX3CR1-SNCA mice”, however Fig. 3I

shows a much stronger effect for DA survival in DA-SNCA compared to CX3CR1-SNCA mice. It would also have been interesting to test, 5 weeks after viral transduction, whether expression of mutated aSYN in both DA neurons and microglia has a synergetic effect on DA neurons death.

Following the reviewer's suggestion, we amended the sentence in line 210 to highlight the higher neurodegeneration occurring in DAT-SNCA respect to CX3CR1-SNCA animals. Moreover, as suggested by the reviewer, we injected wild type animals with an LV vector equipped with a constitutive promoter for diffuse α SYN expression in both neuronal and glial cells. As shown in Sup. Fig. 8, α SYN constitutive expression in the nigral tissue for 5 weeks led to a ~66% reduction of TH+ neurons compared to dGFP injected mice. Thus, neuronal loss resulted twice higher in these mice compared to CX3CR1- and DAT-SNCA animals that exhibited ~32% and ~28% of cell loss, respectively, at the same time point. These observations are in line with a synergistic effect between the α SYN cell-autonomous toxicity in neurons and the detrimental effects triggered by the glial inflammation. However, since in this setting α SYN is also expressed by astrocytes, we cannot exclude an additional contribution of these cells in the neurodegenerative process.

4.Line 232: The authors state that "...only in CX3CR1-SNCA mice the microglia exhibited a strong inflammatory state". This clearly not true. Indeed, hierarchical clustering in supp-Fig 5C and supp-Fig 5D clearly show intermediate inflammatory states in DAT-SNCA mice. Similarly, T-cells infiltration is higher in DAT-SNCA compared to control mice (supp-Fig 6).

Our line of reasoning was that CX3CR1-SNCA respect to DAT-SNCA mice present higher levels of inflammatory markers and infiltrated T cells. Having said that, the reviewer is right indicating that DAT-SNCA mice exhibit intermediate levels of inflammation and T cell infiltration respect to the control and CX3CR1-SNCA conditions.

5.Lines 250-316: It is not clear why the authors did not analyze all the cells from their scRNA-seq experiment together in the same UMAP plot? Presenting the different cell populations on the same plot would make it possible to highlight the cell populations and sub-populations that are common to the two experimental conditions and those that are different. This part of the results is also less clearly presented, the genes mentioned in the text are not always those shown on the figures. As a typical example of this, lines 262 the authors state that they identified Cluster 1-3 as microglial cells because they expressed the cardinal genes *Olfml3*, *Gpr34*, *Siglech* and *Cx3cr1*; and referred to Fig. 4B,C. However, of those only *Cx3cr1* is shown in Fig. 4C. Similarly, Cluster 4 is not clearly defined and with discrepancies between the figures and the text. Overall, this part should be revised to ensure better consistency between the text and the figures. In the figures, identification of the cell populations/subpopulation is not clear, and there are clearly offsets between the colors indicating the different cell populations and the actual names of the populations/subpopulations.

The authors should also compare and comment how the microglial signature they unravel overlaps with other microglia signatures in disease states (i.e. comparison with DAMs, MGnD, ARMs...).

We avoided to analyze the scRNA-seq datasets since the number of total analyzed cells is different between the two conditions, since in the wild type condition the immune cell infiltration is significantly lower respect to the pathological state. Thus, we preferred to represent the plots separated while performing a post-hoc analysis of molecular markers between homologous cell populations. We re-organized the presentation of the gene markers in the text for better reflecting the charts presented in the Figures. Given the high number of genes nominated in the text, we opted to illustrate the most significant genes in the different cell populations for avoiding extremely long

and complex tables difficult to be read. All the datasets will be freely accessible at the time of the publication given the opportunity to further investigate the gene signature of these cell populations for anyone interested.

6. Figure 6: From the text, it seems that effects of LV transduction and α SYN fibrils application was compared, however the second line of the heatmap indicates treatment with IFN γ . The legend of the figure does not shed light the actual conditions compared.

Microglia transduced with the α SYN expressing LV were analyzed for gene expression, P62/ α SYN staining and phagocytosis activity. The results are presented in Fig. 6.

7. Line 528: The authors referred to B6;FVB-Tg(Aldh1l1-cre/ERT2)1Khakh/J as being the CX3CR1-creERT2 mice.... Hopefully this is a copy/paste mistake.... Additionally, in this paragraph of the materials and methods section, the authors should provide the legal agreement number of their animal facility as well as the legal agreement number for their specific project.

We apologize for this evident typo that we amended in the revised version. We also added in the M&M a full description of the legal and ethical agreements covering this study.

8. Line 673: It is not clear why the authors did not use the same type of correction for multiple testing (i.e. they used either Bonferroni or two-stage linear step-up procedure of Benjamini, Krieger and Yekutieli) in the post-hoc tests they performed in ANOVA analyses.

In the revised version we better detailed the statistical analysis. As a matter of fact, we used Anova with groups with normal distribution followed by Bonferroni. In case of non-normal distribution (in some of in vitro experiments) we used the Kurskall Wallis test followed by the two stages step up method.

Minor comments:

1. The authors should comply the official nomenclature when naming mouse & human genes and proteins (DOI: 10.4161/auto.20665)
2. Line 180: The authors should refer to Fig. 2, not Fig 1.
3. Supp figure 5: For readability of the figures, it would be better to keep the same color codes for the experimental groups throughout the manuscript.
4. Line 288: "Chatepsin" should be replaced by "cathepsin"
5. Line 446: the reference is truncated
6. Lines 572-576: This part of the text appears in red...
7. Line 675: A space is missing between "using" and "one"
8. Line 724: "CXCR1" should be replaced by "CX3CR1"
9. Line 727: "n=8" should be replaced by "n=6-8"
10. Line 741: spelling error in "transduction"
11. Line 788: please include all the pvalues shown on the graph
12. Line 857: Panel Supp-Fig5E is not described in the legend of the figure. A diverging scale bar will help appreciate the difference of expression between the different experimental conditions
13. Line 873 & 879: Supp-Fig7 and 8: Insert scale bars on the images
14. Line 886: include the statistics of the graph shown in supp-Fig9

We changed accordingly the text following the reviewer's remarks. In addition, the revised manuscript was further edited by a dedicated commercial service.

Reviewers' Comments:

Reviewer #1:

Remarks to the Author:

The authors have made a good effort trying to address the reviewers comments, which has improved the manuscript significantly.

However, this reviewer still has a major concern regarding the physiological relevance of overexpressing aSYN in microglia. Indeed, all available PD models are to a certain extent artificial, but in this study overexpression of aSYN in microglia most probably does not mimick the human disease. Indeed, the autopsy stainings provided in supp fig 12 and 13 do not show aSYN immunoreactivity in microglia. One would probably achieve a similar effect by overexpressing another aggregation-prone protein in microglia.

A second concern is on the specificity of the effect and the controls. The authors mention that they use dGFP instead of GFP because of the toxicity for DA neurons. This has indeed been reported, although there is no general consensus in the field and several studies have used GFP as control in DA neurons without significant toxicity. However, since the GFP in this study in the control condition is expressed in microglia and not DA neurons, the potential toxicity is not a valid argument. The ideal control would be to have expression of an unrelated protein to similar levels of the protein of interest. The authors have provided supplementary figure 4 to demonstrate similar transgene expression levels, but this is based on qPCR for WPRE. These data demonstrate similar transduction levels, but not similar protein levels. Indeed, gene expression can be regulated differently and/or the stability of the transgenic protein can be different. To be able to compare transgene expression levels, a low magnification overview of immunohistochemical stainings for aSYN and dGFP would be useful. Of course, this still depends on antibody sensitivity, but it can already give an approximate estimation of area and levels of transgene expression. There is still some confusion on the definition of the controls in the text and the figures : the authors mention in their rebuttal that Control means CX3CR1-creERT2 mice injected with the cre-inducible destabilized GFP (dGFP), but this is not clearly mentioned in the different figures and figure legends. For example in Fig 1 control = LV-Flex-dGFP in wild type mice. In figure 7 it seems that control refers to non-injected side. In figure 2a why is the costaining with TH in the control condition and not with Iba-1 if transgene expression is in microglia? Only in figure 4 the control is correctly mentioned as CX3CR1-dGFP, but this should be done consistently in all the figures.

Minor remarks

Supp Fig 3 b-k : is it GFP or dGFP that is used here? Please specify in the figure and figure legend

Supp Fig 6b : the aSYN staining does not seem to be restricted to TH-positive neurons here. Why is that?

Supp Fig 8 : the authors claim that simultaneous aSYN overexpression exacerbates DA neurodegeneration by testing a non-flexed lentiviral vector. However, it is not clear what cell types are transduced here and in what proportion. Most studies with lentiviral vectors in rodent brain using ubiquitous promoters report a majority of neurons transduced (approximately 70-80%) in addition to astrocytes. Did the authors confirm this? Were microglia also transduced with their lentiviral vector?

Supp Fig 9 This is quite surprising since since AAV-WT aSYN has been shown to induce DA neurodegeneration in mice. Did the authors confirm similar expression levels of WT vs mutant aSYN?

Reviewer #2:

Remarks to the Author:

A number of the original concerns have been properly addressed in this revised version of the manuscript. Two important issues should be further considered, however.

1. Two of the Reviewers raised the concern that extrapolation of the results of this study to pathogenetic processes in Parkinson brain requires a note of caution. This is because, for example, no evidence supports the possibility that selective accumulation of alpha-synuclein into microglia represents a primary event triggering nigrostriatal degeneration in PD. This issue should be at least acknowledged in the Discussion section.
2. Another important issues that should be mentioned/discussed in the manuscript is that the extent to which the microglial changes (morphological, functional and toxic) described in this paper are "specific" to alpha-synuclein accumulation into these cells remains unclear.

Reviewer #3:

Remarks to the Author:

The authors of the manuscript «Microglia-specific overexpression of alpha-Synuclein leads to severe dopaminergic neurodegeneration by phagocytic exhaustion and oxidative toxicity» have answered most of the question raised in the initial review, they also performed new quantifications and performed additional experiments that significantly reinforced the strength and the relevance of the work.

However, some points still need to be further improved, with the main concern being their presentation of their scRNA-seq results. Indeed, in their rebuttal letter, the authors state that they did not analyzed datasets from the two conditions because the number of cells was different between the conditions. Actually, having different number of cells between the two conditions is not an issue as far as the sequencing depth is, in average, the same for cells in all conditions. In the main text, the authors report an average sequencing depth of 1554 transcript per cell (of note sequencing depth should be given as number of reads per cell not number of transcripts per cell), not mentioning any differences between CX3CR1-dGFP and CX3CR1-SNCA conditions. Should the sequencing depth be different between the conditions, the authors still can normalize their data so that the sequencing depth will not be different between the two conditions. Post-hoc comparisons of the cell populations in the two different experimental conditions is neither a standardized nor a good approach to compare cell populations in two different conditions. In particular, it cannot be understood from the results presented in Figure 4 and Figure 5 how microglia A1/A2/A3 identified in CX3CR1-SNCA related to microglia microglia 1/2/A3 CX3CR1-dGFP. The same apply to CD8 cells.

In addition to these fundamental aspects of scRNA-seq data analysis, the authors did not modify the Figures 4 and 5 to make them easier to read and to understand (they only changed the main text). Among the points that render the figures difficult to read are:

1. offsets between the colors indicating the different cell populations and the actual names of the populations/subpopulations (circled in red in the image joined of this review)
2. In figure 4C, labelling of Mast cells and APCs seem to be inverted in Fig 4C

Considering the general purpose of their work (i.e. underlining the contribution of microglia to PD pathogenesis), it is also unfortunate that the authors did not show how the microglial gene signature they identified in CX3CR1-SNCA conditions overlaps with previously identified microglia gene signature identified in neurodegenerative conditions (i.e. DAMs, MGnD, etc.).

Regarding this part, the designation "Immune infiltrate" does seem adequate as this analysis also refer to microglia which are brain resident microglia and not infiltrates.

Other more minor comments:

- Overaccumulation of aSyn: Using qPCRs, the authors now show that dGfp and Syn transgene express at the same level. However, they did not quantify the level of overexpression compared to the endogeneous aSyn.
- Effects of lentivirus injection: The authors claim that Figure 2 and 3 show that after lentiviral transduction with dGFP there is no sign of inflammation. However, their analysis is only based on morphological analysis and CD68 labelling which are very crude analysis. Additionally, quantification in Figure sup 3 shows slightly higher percentage of CD68+ surface (and to a lesser extend higher IBA1+ surface) in CX3CR1-dGFP compared to saline treated mice.
- Regarding microglia morphology analysis, how fractal analysis was performed is not described in the materials and methods section.
- We appreciated that the authors tested whether aSYN overexpression in microglia, neurons and astrocytes had synergetic effect on DA neuron loss (Figure supp8). As, they now quantified DA loss

in VTA and SNc in CX3CR1-dGFP and CX3CR1-SNCA conditions (Figure 3), they should precise to which region the quantification shown in Figure supp8 relates, SNc, VTA or both.

- In their rebuttal letter, the authors agree on the fact that DAT-SNCA mice display an intermediate inflammatory state compared to CX3CR1-SNCA mice, however they did not modify their sentence in the main text. Their sentence "...microglia exhibited robust inflammation only in CX3CR1-SNCA" is somehow tendentious and should be rewritten to stick to the actual results.
- The authors corrected the reference to B6;FVB-Tg(Aldh1l1-cre/ERT2)1Khakh/J mice in the main text but not in the Nature reporting summary (doc 284837_1_supp_0_qsqd5l.pdf).

REVIEWER COMMENTS

Reviewer #1 (Remarks to the Author):

The authors have made a good effort trying to address the reviewers comments, which has improved the manuscript significantly.

However, this reviewer still has a major concern regarding the physiological relevance of overexpressing aSYN in microglia. Indeed, all available PD models are to a certain extent artificial, but in this study overexpression of aSYN in microglia most probably does not mimick the human disease. Indeed, the autopsy stainings provided in supp fig 12 and 13 do not show aSYN immunoreactivity in microglia. One would probably achieve a similar effect by overexpressing another aggregation-prone protein in microglia.

As remarked previously, our study established an *in vivo* model where it is possible to identify the specific role played by microglia, once intoxicated by α Syn accumulation, in the pathophysiological mechanisms that lead to neurodegeneration. We removed any assumption to pretend this as a model of human PD, yet we emphasized the development of a powerful system which allowed us to extrapolate the pathological contribution of α Syn accumulation selectively in microglia. Thus, we uncovered new aspects of inflammation that impinge on the neurodegenerative process in the brain, which might be of particular interest in at least three major aspects:

- 1- Even though there is no clear description of α Syn accumulation in microglia in PD neuropathology studies, this cell type senses early on the presence of α Syn aggregates in the parenchyma and gets activated by enhancing its phagocytic activity and release of pro-inflammatory cytokines (Choi et al., Nature Comm 2020; Olanow et al. Brain 2019).
- 2- Protracted microglia activation might become detrimental for neuronal survival, in particular for those neurons that are particularly sensitive to noxious stimuli like the DAergic ones.
- 3- In PD, important genetic risk factors involved genes encoding for lysosomal proteins that are highly expressed in microglia. Indeed, there are evidence that lysosomal defects can significantly affect microglia behaviour in models of PD (review by Tremblay et al., Mol Neurodege. 2019).

As speculated by the reviewer, it is possible that the noxious effects triggered by microglia once accumulating α Syn can be recapitulated by the accumulation of other aggregation-prone proteins. However, we provided solid evidence that this does not occur when microglia are expressing either GFP or its destabilized form at similar levels (Suppl. Figure 3). Thus, these findings open the intriguing possibility that microglia might be particularly liable to accumulation of other aggregation-prone proteins, such as TAU and TDP-43 that play major roles in the onset of other neurodegenerative disorders. We commented this point in the discussion.

A second concern is on the specificity of the effect and the controls. The authors mention that they use dGFP instead of GFP because of the toxicity for DA neurons. This has indeed been reported, although there is no general consensus in the field and several studies have used GFP as control in DA neurons without significant toxicity. However, since the GFP in this study in the control condition is expressed in microglia and not DA neurons, the potential toxicity is not a valid argument. The ideal control would be to have expression of an unrelated protein to similar levels of the protein of interest. The authors have provided supplementary figure 4 to demonstrate similar transgene expression levels, but this is based on qPCR for WPRE. These data demonstrate

similar transduction levels, but not similar protein levels. Indeed, gene expression can be regulated differently and/or the stability of the transgenic protein can be different. To be able to compare transgene expression levels, a low magnification overview of immunohistochemical stainings for aSYN and dGFP would be useful. Of course, this still depends on antibody sensitivity, but it can already give an approximate estimation of area and levels of transgene expression.

To address this request, we performed immunohistochemistry for α SYN and GFP on CX3CR1-SNCA and CX3CR1-dGFP nigral tissues, respectively. In the new Suppl. Figure 4, we included representative low magnification pictures of immunohistochemistry for α SYN and dGFP showing that the distribution and intensity of signals are extremely comparable between the two conditions.

To further expand on the comment raised by the reviewer, we also compared the levels of microgliosis triggered by the expression of dGFP (destabilized) or native GFP. As we observed for the dGFP, the accumulation of native GFP in microglia does not provoke evident signs of aberrant activation as highlighted by IBA1 and CD68 staining (Suppl. Figure 3b-k). These results indicate that GFP does not provoke notable toxic effects in microglia as already shown for the destabilized form of GFP. Nonetheless, we privileged the use of dGFP in this work, since we presented strong evidence that native GFP expression in DAergic neurons have substantial toxic effects (Suppl. Figure 3a). This last piece of data might indicate that the clearance machinery of microglia is highly efficient, sustaining a more efficient handling of protein accumulation respect to neurons.

There is still some confusion on the definition of the controls in the text and the figures : the authors mention in their rebuttal that Control means CX3CR1-creERT2 mice injected with the cre-inducible destabilized GFP (dGFP), but this is not clearly mentioned in the different figures and figure legends. For example in Fig 1 control = LV-Flex-dGFP in wild type mice. In figure 7 it seems that control refers to non-injected side. In figure 2a why is the costaining with TH in the control condition and not with Iba-1 if transgene expression is in microglia? Only in figure 4 the control is correctly mentioned as CX3CR1-dGFP, but this should be done consistently in all the figures.

We made explicit for each figure and its relative legend which control condition was used. Figure 2A shows a double immunofluorescence for pS129 α Syn and TH in the control condition showing that no signal is detectable for the phosphorylated form of α Syn as expected. We opted to maintain this pattern to keep the order for each staining within the same column of the figure that should facilitate the reading and consistency of the figure.

Minor remarks:

Supp Fig 3 b-k : is it GFP or dGFP that is used here? Please specify in the figure and figure legend

This was GFP in order to highlight that native GFP expression does not elicit aberrant activation of microglia similar to what we previously showed for the destabilized form of GFP (dGFP).

Supp Fig 6b : the aSYN staining does not seem to be restricted to TH-positive neurons here. Why is that?

We thank the reviewer to note this mild difference. This is not representative of our findings where we constantly detected viral SNCA expression restricted to microglial cells in CX3CR1-SNCA

mice even after 15 weeks from viral transduction. Thus, we added new images that better illustrate our findings.

Supp Fig 8 : the authors claim that simultaneous aSYN overexpression exacerbates DA neurodegeneration by testing a non-flexed lentiviral vector. However, it is not clear what cell types are transduced here and in what proportion. Most studies with lentiviral vectors in rodent brain using ubiquitous promoters report a majority of neurons transduced (approximately 70-80%) in addition to astrocytes. Did the authors confirm this? Were microglia also transduced with their lentiviral vector?

To answer to this comment, we performed double immunostaining for the viral human SNCA and selective neuronal and glial markers. As anticipated by the reviewer, the lentiviral transduction targeted preferentially neuronal cells. However, both astrocytes and microglia were expressing the viral transgene although in a minor fraction. These new data including the quantitative analysis are presented in Suppl. Figure 8.

Supp Fig 9 This is quite surprising since since AAV-WT aSYN has been shown to induce DA neurodegeneration in mice. Did the authors confirm similar expression levels of WT vs mutant aSYN?

Indeed, AAV based expression of α SYN induces DAergic neurodegeneration, but the extent of neuronal loss over time strictly depends by multiple factors among which the choice of the promoter and in which cell types α SYN is expressed beyond neurons. In Suppl. Fig. 9, we compared the effects of expressing wild type SNCA either in DAergic neurons or in microglia. This restricted pattern of expression is not normally used to induce a fast neurodegeneration which is achievable only by constitutive SNCA expression (i.e. Decressac et al., Neurobiol. Dis 2012). Nonetheless, also DAT-SNCA mice showed a clear trend of DAergic neuronal loss at 5 weeks, which would become probably statistically significant after some more weeks from the viral transduction. To answer to the reviewer's request, we performed immunohistochemistry to assess the α SYN protein levels sustained by the viral vector expressing either the wild-type or mutated (G420A) SNCA. As shown in Suppl. Figure 4d,e, α SYN protein levels were comparable using the two different SNCA forms on nigral tissue sections.

Reviewer #2 (Remarks to the Author):

A number of the original concerns have been properly addressed in this revised version of the manuscript. Two important issues should be further considered, however.

1. Two of the Reviewers raised the concern that extrapolation of the results of this study to pathogenetic processes in Parkinson brain requires a note of caution. This is because, for example, no evidence supports the possibility that selective accumulation of alpha-synuclein into microglia represents a primary event triggering nigrostriatal degeneration in PD. This issue should be at least acknowledged in the Discussion section.

We fully agree with this comment of the reviewer. Indeed, even though there is no clear description of α Syn accumulation in microglia in PD neuropathology studies, this cell type senses early on the presence of α Syn aggregates in the parenchyma and gets activated by altering its

phagocytic activity and enhancing its release of pro-inflammatory cytokines that represent an important pathological aspect that endangers neuronal survival (Choi et al., Nature Comm 2020; Olanow et al. Brain 2019). Thus, we added this comment in the discussion at page 16: *“CX3CR1-SNCA mice are not a comprehensive model for PD since α SYN accumulation in microglia is not a first etiopathological event in human conditions. However, these animals represent a unique system in which the responses of either the neurons or the adaptive immune system exerted by the α SYN intoxicated microglia can be selectively investigated to decipher the underlying disease mechanisms and identify and validate new immune-mediated translational approaches.”*

2. Another important issue that should be mentioned/discussed in the manuscript is that the extent to which the microglial changes (morphological, functional and toxic) described in this paper are “specific” to alpha-synuclein accumulation into these cells remains unclear.

In line with this comment of the reviewer we included an additional sentence in the discussion exactly on this argument. Indeed, our study provides strong evidence that accumulation of α Syn, but not the native GFP or its destabilized form, triggers a chain of detrimental events which culminate with infiltration of peripheral immune cells, microglia exhaustion and production of toxic molecules. These findings might suggest the intriguing possibility that aggregation-prone proteins might be particularly harmful to microglia suggesting that other disease-associated proteins like TAU and TDP-43 can share similar effects in this context. Thus, we added the following comment in the discussion at page 17: *“Herein, we unveiled the neurotoxic effects of microglia accumulating α SYN, but not native GFP or its destabilized form. These findings raise the intriguing possibility that, beyond α SYN, other aggregation-prone proteins, like TAU and TDP-43, might share a similar neuropathological mechanism contributing to the progression of other relevant neurodegenerative diseases.”*

Reviewer #3 (Remarks to the Author):

The authors of the manuscript «Microglia-specific overexpression of alpha-Synuclein leads to severe dopaminergic neurodegeneration by phagocytic exhaustion and oxidative toxicity» have answered most of the question raised in the initial review, they also performed new quantifications and performed additional experiments that significantly reinforced the strength and the relevance of the work.

However, some points still need to be further improved, with the main concern being their presentation of their scRNA-seq results. Indeed, in their rebuttal letter, the authors state that they did not analyze datasets from the two conditions because the number of cells was different between the conditions. Actually, having a different number of cells between the two conditions is not an issue as far as the sequencing depth is, in average, the same for cells in all conditions. In the main text, the authors report an average sequencing depth of 1554 transcript per cell (of note sequencing depth should be given as number of reads per cell not number of transcripts per cell), not mentioning any differences between CX3CR1-dGFP and CX3CR1-SNCA conditions. Should the sequencing depth be different between the conditions, the authors still can normalize their data so that the sequencing depth will not be different between the two conditions. Post-hoc comparisons of the cell populations in the two different experimental conditions is neither a standardized nor a good approach to compare cell populations in two different conditions. In

particular, it cannot be understood from the results presented in Figure 4 and Figure 5 how microglia A1/A2/A3 identified in CX3CR1-SNCA related to microglia microglia 1/2/A3 CX3CR1-dGFP. The same apply to CD8 cells.

In the original manuscript the reads for cell were reported in the methods section where we wrote *"Libraries were sequenced on a Novaseq 6000 flowcell (Illumina), with a minimum depth of 50K reads/cell."* However, we agree with the general comment of the reviewer and, following his/her suggestions, we modified the computational analysis by integrating the datasets between the CX3CR1-dGFP and CX3CR1-SNCA conditions. Even after this computational elaboration, the results were mostly comparable with those presented earlier both in terms of cell clustering and definition of the activation state of the different cell populations. The only exception is that with this integrated analysis, the cluster originally corresponding to the monocytes was merged within the microglial populations. This might be expected since the two populations are very alike, and their similarity is even greater in a highly inflammatory condition. We acknowledged this caveat in the presentation of the results. Except this, all our findings and conclusions are equally supported after this additional elaboration. As suggested by the reviewer, this new analysis enabled an easy visualization of the difference in cell composition in each cluster between the CX3CR1-dGFP and CX3CR1-SNCA conditions as we highlighted in new Figure 4b.

We described in full the computational pipeline used for this analysis in the revised Methods as following: *"We processed raw paired-end scRNA-seq data using cellRanger (10x Genomics) with default parameters to generate the DGE matrices (Zheng et al., 2017). We performed alignment to the mm10 reference genome using cellRanger "count" with Gencode v37 annotations. We achieved unique mapping for around 90% of the reads both in dGFP in SNCA conditions. We discarded non-uniquely mapped reads. To distinguish between those captured cellular transcriptomes from those that captured ambient RNA, we sorted barcodes by decreasing number of reads and picked the inflection point ('knee') of the cumulative fraction of reads plot. We used Seurat 4 for downstream computational analyses (Yuhan et al., 2020). The two datasets were integrated first finding the anchors shared between them using the "FindIntegrationAnchors" function followed by IntegrateData merging. To remove damaged cells, we extracted the percentage of mitochondrial reads and the count of captured transcripts (nCount_RNA) and removed all barcodes with < 200 nCount_RNA or high-percentage of mitochondrial reads (>20 % in both d-GFP and SNCA). We removed barcodes with extremely low mitochondrial reads (<0.8%) to exclude nuclei. In order to exclude potential doublets, we also excluded cells with very high nUMI (>5000). We were able to analyze 696 and 3273 cells for dGFP and SNCA conditions, respectively. For each cell, UMI counts per gene were normalized and scaled. We performed clustering considering only the top 2000 highly variable genes, as identified by the function "FindVariableGenes". Variable genes were then used to perform principal component (PC) analysis. We selected the PCs to be used for downstream analyses by evaluating the "PCel-bowPlot" and the "JackStrawPlot". We used the first 20 PCs for both dGFP and SNCA conditions. We identified clusters using the function "FindClusters", which exploits a SNN modularity optimization clustering algorithm (at Resolution=0.5). We visualized clusters using the uniform manifold approximation and projection (UMAP) dimensionality reduction. We used the manual inspection of marker genes determined using the "FindAllMarkers" function for cluster identification. This function determines which genes, that are expressed in at least three cells, are enriched in every clustering using log₂FC threshold values of 0.25 and 0.05 of adjusted pValue (FDR)."*

In addition to these fundamental aspects of scRNA-seq data analysis, the authors did not modify the Figures 4 and 5 to make them easier to read and to understand (they only changed the main text). Among the points that render the figures difficult to read are:

1. offsets between the colors indicating the different cell populations and the actual names of the populations/subpopulations (circled in red in the image joined of this review)
2. In figure 4C, labelling of Mast cells and APCs seem to be inverted in Fig 4C

Considering the general purpose of their work (i.e. underlining the contribution of microglia to PD pathogenesis), it is also unfortunate that the authors did not show how the microglial gene signature they identified in CX3CR1-SNCA conditions overlaps with previously identified microglia gene signature identified in neurodegenerative conditions (i.e. DAMs, MGnD, etc.).

Regarding this part, the designation “Immune infiltrate” does seem adequate as this analysis also refer to microglia which are brain resident microglia and not infiltrates.

We modified Figures 4 and 5 reflecting the new computational analysis and the illustrations were partially amended changing colors and adding numbers to the UMAP plots for a straightforward identification of the cellular clusters. We did not perform a direct comparison with microglia of other neurodegenerative conditions since we plan a more extensive analysis of this subject which is requiring a substantial effort and time for a next follow up of the study. We eliminated the definition of “immune infiltrate” as rightly pointed out by the reviewer.

Other more minor comments:

- Overaccumulation of aSyn: Using qPCRs, the authors now show that dGfp and Syn transgene express at the same level. However, they did not quantify the level of overexpression compared to the endogenous aSyn.

We extended the analysis of the viral transgenes showing that also protein levels of dGFP and human α SYN are comparable as detected by immunohistochemical analysis on CX3CR1-SNCA and CX3CR1-dGFP nigral tissues (Suppl. Figure 4d,e). However, we could not provide a direct comparison of the viral and endogenous α SYN protein levels for the following reasons. First, we expressed α SYN in selected cell populations as in microglia and DAergic neurons in CX3CR1-dGFP and CX3CR1-SNCA mice, respectively, and therefore a Western blotting on total nigral tissue lysates would not be informative by itself. Second, our work is mainly focused on viral α SYN expression in microglia where in a normal condition endogenous α SYN levels are extremely low and hard to be detectable by standard Western blotting. In fact, our overall paradigm and rationale are that only during PD disease progression microglial cells accumulate exogenous α SYN which is released by affected neurons. Thus, in this study we modeled this scenario by overexpressing α SYN selectively into the microglia. Given this background, it remains less informative to give levels of endogenous α SYN in microglia in a normal condition.

- Effects of lentivirus injection: The authors claim that Figure 2 and 3 show that after lentiviral transduction with dGFP there is no sign of inflammation. However, their analysis is only based on morphological analysis and CD68 labelling which are very crude analysis. Additionally, quantification in Figure sup 3 shows slightly higher percentage of CD68+ surface (and to a lesser extend higher IBA1+ surface) in CX3CR1-dGFP compared to saline treated mice.

In this study morphological analysis and CD68 labeling are used as entry point to determine the microglia state in the different genetic conditions. This represents only an initial characterization that is then followed by extensive genomic analyses that we performed by bulk and single-cell

RNA-seq. The molecular analysis further confirmed and extended the findings initially reported by morphological and histochemical methods.

- Regarding microglia morphology analysis, how fractal analysis was performed is not described in the materials and methods section.

We added this new paragraph in the Methods describing the methodology used to perform fractal analysis: *“The features of microglia morphology were elaborated by FraCLac tool of Image J software. Briefly, the immunofluorescent staining for Iba1 was acquired with 40x magnification lens and each single microglial cell was isolated from the context. The resulting image (containing one cell) was submitted to FraCLac for the automatic detection of the Convex Hull and the Bounding Circle edges, together with the number of total and foreground pixels. The information was integrated to provide different indices recapitulating the cell shape, including the fractal dimension which represents a statistical index of the shape complexity and of the space-filling capacity of a pattern.”*

- We appreciated that the authors tested whether aSYN overexpression in microglia, neurons and astrocytes had synergetic effect on DA neuron loss (Figure supp8). As, they now quantified DA loss in VTA and SNc in CX3CR1-dGFP and CX3CR1-SNCA conditions (Figure 3), they should precise to which region the quantification shown in Figure supp8 relates, SNc, VTA or both.

Graph in Figure Supp. 8a reports the DA neuronal loss in the *substantia nigra*. We added this information in the revised Figure legend.

- In their rebuttal letter, the authors agree on the fact that DAT-SNCA mice display an intermediate inflammatory state compared to CX3CR1-SNCA mice, however they did not modify their sentence in the main text. Their sentence “...microglia exhibited robust inflammation only in CX3CR1-SNCA” is somehow tendentious and should be rewritten to stick to the actual results.

We rephrased the sentence as following: *“Taken together, these findings provide evidence that microglia exhibited the highest inflammation levels in CX3CR1-SNCA mice, and this effect was likely caused by the cell-autonomous accumulation of α SYN.”* We don't want to deny the presence of inflammation in the DAT-SNCA mice that certainly is present, but already known. Our focus is to highlight the novel findings related to the CX3CR1-SNCA animals that certainly have the highest inflammatory state.

- The authors corrected the reference to B6;FVB-Tg(Aldh1l1-cre/ERT2)1Khakh/J mice in the main text but not in the Nature reporting summary (doc 284837_1_supp_0_qsqa5l.pdf).

We amended the Nature Reporting Summary accordingly.

Reviewers' Comments:

Reviewer #1:

Remarks to the Author:

The authors have adequately addressed most comments, except for the concern on the physiological relevance and specificity. The authors claim that they wanted to study pathological contribution of α Syn accumulation selectively in microglia. Microglia might indeed take up extracellular α Syn released from neurons, but this situation is quite different from inducing microglia to overexpress α Syn. This would be mainly soluble cytoplasmic α Syn, while in the other situation mostly aggregated protein in the endolysosomal compartment.

Minor comment :

Supp Fig 6b : the α SYN staining does not seem to be restricted to TH-positive neurons here. Why is that?

We thank the reviewer to note this mild difference. This is not representative of our findings where we constantly detected viral SNCA expression restricted to microglial cells in CX3CR1-SNCA mice even after 15 weeks from viral transduction. Thus, we added new images that better illustrate our findings.

The question was actually not referring to the CX3CR1-SNCA condition, but to the DAT-SNCA condition, where α SYN staining is not restricted to DA cells, but seems to also colocalize significantly with Iba1 staining.

Reviewer #2:

Remarks to the Author:

The Authors were asked to discuss important caveats underlying this study, particularly in relation to the pathophysiological relevance of their experimental model and the specificity of the results that were obtained after pronounced and sustained α SYN accumulation within microglia. In response to this suggestion, a new sentence was added indicating "CX3CR1-SNCA mice are not a comprehensive model for PD since α SYN accumulation in microglia is not a first etiopathological event in the human condition". The issue of relevance of the work in the context of a human disease probably deserves greater attention. Moreover, the added sentence is relatively superficial and potentially misleading. Pronounced and sustained α SYN accumulation within microglia (as induced in the mouse model described in this study) does not mimic any PD pathological feature and is not a feature at either early or later disease stages. It is a novel experimental paradigm in which transgenic α SYN accumulation was (probably) intended to exacerbate α SYN-microglia interactions and to magnify toxic and pathological consequences of these interactions. With this important caveat in mind, it is possible that this model may help investigate mechanisms bearing pathogenetic implications, as suggested by the Authors in the last paragraph of the Discussion section.

Reviewer #3:

Remarks to the Author:

A large number of the issues raised by the reviewers have now been addressed by the authors. In particular, as suggested by this reviewer, they revised the format of their scRNA-seq data analysis and integrated the data obtained in the CX3CR1-dGFP and CX3CR1-SNCA conditions into the same analysis. In line with this new analysis of their data, they considerably revised figures 4 and 5 of their manuscript. However further improvements could be made to ease the comprehension of their results.

In particular, in Figure 4B - lower part, the authors show, for each of the cluster identified, the fraction of cells from either the CX3CR1-dGFP or the CX3CR1-SNCA conditions. This analysis is interesting but as the number of cells injected in the analysis differs between the two conditions (i.e. respectively 636 and 3273 for the CX3CR1-dGFP and the CX3CR1-SNCA conditions), this representation is, as such, misleading with all the clusters appearing enriched for cells of the CX3CR1-SNCA condition. It would be more informative to present the data as the percentage of cells for each condition in each cluster. In addition, by showing in each condition, the percentage

of cells from the different clusters, the more complex composition of innate and adaptive immune cells in the CX3CR1-SNCA condition will become clearer.

Does figure 5 only refers to counts in CX3CR1-SNCA conditions as suggested by the headband at the top of the figure? If so, this is (1) not consistent with the new analysis of the scRNA-seq data and (2) not mentioned in the legend of the figure.

Another confusing point is the mention by the authors of the use of "FLEX-AAV for the conditional expression of the wild type α SYN which was injected in CX3CR1-creERT2 mice for selective expression in microglia" (lines 235-242) whereas they show and state earlier that "high specificity of the FLEX cassette conditional system was only achieved when the system was packaged in lentiviruses" (lines 162-163).

Minor points:

- The number of cells of each condition included in the analysis could be indicated in Figure 4b - upper part.
- There are still some offset issues between the columns and their naming, particularly in panels figure 5, panels b and c.
- In their rebuttal letter, the authors state that they eliminated the term "immune infiltrate". However, the term still appears at different places: lines 137, 348 and 512.
- In the legend of Figure-supp-8 and at line 731, IBA1 refers to the protein and should be capitalized.
- In the main text (lines 271-272), the authors refer to several T-cell markers, in particular Cd161 but in Figure-supp-11 they report expression level for Cd160.

Reviewer #1 (Remarks to the Author):

The authors have adequately addressed most comments, except for the concern on the physiological relevance and specificity. The authors claim that they wanted to study pathological contribution of α Syn accumulation selectively in microglia. Microglia might indeed take up extracellular α Syn released from neurons, but this situation is quite different from inducing microglia to overexpress α Syn. This would be mainly soluble cytoplasmic α Syn, while in the other situation mostly aggregated protein in the endolysosomal compartment.

We thank the reviewer for this comment that we have elaborated in the discussion, as following: “CX3CR1-SNCA mice are not a comprehensive model for PD since α SYN accumulation in microglia is not a first etiopathological event in the human condition. In addition, in the human disease the microglia accumulate mainly extracellular α SYN released from neurons. However, herein we forced SNCA expression in microglial cells in order to accelerate α SYN accumulation and exacerbate its dependent microglial response. Thus, these animals represent a unique system in which the responses by either the neurons or the adaptive immune system exerted to the α SYN intoxicated microglia can be selectively investigated to decipher the underlying disease mechanisms and identify and validate new immune-mediated translational approaches.”

Minor comment :

Supp Fig 6b : the α SYN staining does not seem to be restricted to TH-positive neurons here. Why is that?

We thank the reviewer to note this mild difference. This is not representative of our findings where we constantly detected viral SNCA expression restricted to microglial cells in CX3CR1-SNCA mice even after 15 weeks from viral transduction. Thus, we added new images that better illustrate our findings.

The question was actually not referring to the CX3CR1-SNCA condition, but to the DAT-SNCA condition, where α SYN staining is not restricted to DA cells, but seems to also colocalize significantly with Iba1 staining.

As remarked by the reviewer, for the DAT-SNCA condition it is possible to score α SYN staining also in some IBA1+ cells. This is notable only at 15 weeks, but not at the earlier time point of 5 weeks (Suppl. Fig. 6a,b). At 15 weeks DAT-SNCA mice show extensive degeneration of DA neurons in the substantia nigra. Thus, it is likely that α SYN aggregated species have been released by suffering and dying neurons and actively taken up by the surrounding IBA1+ cells. Supporting this hypothesis, the specificity of the SNCA expression in DA neurons is confirmed by our extensive preliminary studies (Suppl. Fig 1k) and the selective α SYN localization in DA neurons at the 5 week time-point.

Reviewer #2 (Remarks to the Author):

The Authors were asked to discuss important caveats underlying this study, particularly in relation to the pathophysiological relevance of their experimental model and the specificity of the results that were obtained after pronounced and sustained α SYN accumulation within microglia. In response to this suggestion, a new sentence was added indicating “CX3CR1-SNCA mice are not a comprehensive model for PD since α SYN accumulation in microglia is not a first etiopathological event in the human condition”. The issue of relevance of the work in the context of a human disease probably deserves greater attention. Moreover, the added sentence is relatively superficial and potentially misleading. Pronounced and sustained α SYN accumulation within microglia (as induced in the mouse model described in this study) does not mimic any PD pathological feature and is not a feature at either early or later disease stages. It is a novel experimental paradigm in which

transgenic α SYN accumulation was (probably) intended to exacerbate α SYN-microglia interactions and to magnify toxic and pathological consequences of these interactions. With this important caveat in mind, it is possible that this model may help investigate mechanisms bearing pathogenetic implications, as suggested by the Authors in the last paragraph of the Discussion section.

We thank the reviewer for this comment. However, we do not fully agree with this remark since mounting evidence from different experimental models has recently showed that extracellular α SYN can accumulate in microglial cells, although not detectable as frank aggregates, but sufficient to lead to their activation and induce a detrimental inflammatory response (Lee et al., *Biochem. Biophys. Res. Commun.* 372, 423–428, 2008; Lee et al., *J. Biol. Chem.* 285, 9262–9272, 2010; Kim et al., *Nat. Commun.* 4:1562, 2013; Choi et al., *Nat. Comm* 11:1386, 2020; Gentzel et al., *Neurobiol Aging*. 106:12-25, 2021; Xia et al., *Brain*. awab122, 2021. doi: 10.1093/brain/awab122).

We have further elaborated the discussion as following:

CX3CR1-SNCA mice are not a comprehensive model for PD since α SYN accumulation in microglia is not a first etiopathological event in the human condition. In addition, in the human disease the microglia accumulate mainly extracellular α SYN released from neurons. However, herein we forced SNCA expression in microglial cells in order to accelerate α SYN accumulation and exacerbate its dependent microglial response. Thus, these animals represent a unique system in which the responses by either the neurons or the adaptive immune system exerted to the α SYN intoxicated microglia can be selectively investigated to decipher the underlying disease mechanisms and identify and validate new immune-mediated translational approaches.“

Reviewer #3 (Remarks to the Author):

A large number of the issues raised by the reviewers have now been addressed by the authors. In particular, as suggested by this reviewer, they revised the format of their scRNA-seq data analysis and integrated the data obtained in the CX3CR1-dGFP and CX3CR1-SNCA conditions into the same analysis. In line with this new analysis of their data, they considerably revised figures 4 and 5 of their manuscript. However further improvements could be made to ease the comprehension of their results.

In particular, in Figure 4B - lower part, the authors show, for each of the cluster identified, the fraction of cells from either the CX3CR1-dGFP or the CX3CR1-SNCA conditions. This analysis is interesting but as the number of cells injected in the analysis differs between the two conditions (i.e. respectively 636 and 3273 for the CX3CR1-dGFP and the CX3CR1-SNCA conditions), this representation is, as such, misleading with all the clusters appearing enriched for cells of the CX3CR1-SNCA condition. It would be more informative to present the data as the percentage of cells for each condition in each cluster. In addition, by showing in each condition, the percentage of cells from the different clusters, the more complex composition of innate and adaptive immune cells in the CX3CR1-SNCA condition will become clearer.

As suggested by the reviewer, we replaced the cluster analysis showing the data as a percentage of cells for each cluster in the different conditions (dGFP and SNCA) (Fig. 4b).

Does figure 5 only refers to counts in CX3CR1-SNCA conditions as suggested by the headband at the top of the figure? If so, this is (1) not consistent with the new analysis of the scRNA-seq data and (2) not mentioned in the legend of the figure.

This was a leftover of the previous version. Indeed, the heatmaps illustrated in Fig. 5 have been re-elaborated after the integration of the datasets between the CX3CR1-dGFP and CX3CR1-SNCA

conditions. We have removed this headband in the revised figure.

Another confusing point is the mention by the authors of the use of “FLEX-AAV for the conditional expression of the wild type α SYN which was injected in CX3CR1-creERT2 mice for selective expression in microglia” (lines 235-242) whereas they show and state earlier that “high specificity of the FLEX cassette conditional system was only achieved when the system was packaged in lentiviruses” (lines 162-163).

We thank the reviewer to spot this typo. Indeed, we continued to employ the FLEX-LV vector also for these experiments.

Minor points:

- The number of cells of each condition included in the analysis could be indicated in Figure 4b - upper part.
- There are still some offset issues between the columns and their naming, particularly in panels figure 5, panels b and c.
- In their rebuttal letter, the authors state that they eliminated the term “immune infiltrate”. However, the term still appears at different places: lines 137, 348 and 512.
- In the legend of Figure-supp-8 and at line 731, IBA1 refers to the protein and should be capitalized.
- In the main text (lines 271-272), the authors refer to several T-cell markers, in particular Cd161 but in Figure-supp-11 they report expression level for Cd160.

We changed Fig. 4 and 5 and amended the text according to these remarks of the reviewer (changes highlighted in red).

Reviewers' Comments:

Reviewer #1:

Remarks to the Author:

It might seem a matter of semantics, but the following paragraph provided by the authors 'Thus, these animals represent a unique system in which the responses by either the neurons or the adaptive immune system exerted to the α SYN intoxicated microglia can be selectively investigated to decipher the underlying disease mechanisms and identify and validate new immune-mediated translational approaches' clearly mentions disease mechanisms and translational impact. This is misleading since there is no evidence at all that this corresponds to a disease.

Reviewer #2:

Remarks to the Author:

In the revised sentence added to the Discussion, the Authors now correctly emphasize the fact that microglia may accumulate "extracellular" alpha-synuclein under pathophysiological conditions in PD. They also indicate that this is a significant difference compared with the experimental model described in their study. Whether microglial activation would be comparable when induced by clearance/accumulation of extracellular alpha-synuclein or after constitutive alpha-synuclein overexpression remains unclear and could have probably been further discussed. Nonetheless, the (brief) discussion of caveats concerning the relevance of this model to pathogenetic processes in PD is now acceptable.

Reviewer #3:

Remarks to the Author:

The authors have now addressed the questions raised by the reviewers. The data presented in the manuscript provide valuable findings that would be of interest to researchers studying the role of microglia in Parkinson disease, but also in other neurodegenerative diseases.

Minor points:

- Relating the protein, IBA1 still appears in lower case in several part of different figure (mostly in -y-axis of graph). It should be capitalized. Ex: Fig 2g, 7a; Fig-supp 2,3,6,8
- At reverse, some genes in heatmaps are capitalized when they should be in lower case Ex: Fig 5b (CD206, TGF, IL1, IL6, CCL2), 6a (CD22)

REVIEWERS' COMMENTS

Reviewer #1 (Remarks to the Author):

It might seem a matter of semantics, but the following paragraph provided by the authors 'Thus, these animals represent a unique system in which the responses by either the neurons or the adaptive immune system exerted to the α SYN intoxicated microglia can be selectively investigated to decipher the underlying disease mechanisms and identify and validate new immune-mediated translational approaches' clearly mentions disease mechanisms and translational impact. This is misleading since there is no evidence at all that this corresponds to a disease.

We respectfully disagree with this particular comment of the reviewer. Indeed, in our sentence we paid particular attention to not imply in any means that our mouse “is a new model for human PD”, but on contrary we highlighted that it recapitulates only some of the disease mechanisms (e.g. selective death of DA neurons due to neuroinflammation) that, however, can be specifically exploited for further validation of therapies (e.g. immune-mediated therapies to curb neuroinflammation and peripheral immune cell infiltration). In this respect, we believe this description faithfully recapitulates the significance of our findings.

Reviewer #2 (Remarks to the Author):

In the revised sentence added to the Discussion, the Authors now correctly emphasize the fact that microglia may accumulate "extracellular" alpha-synuclein under pathophysiological conditions in PD. They also indicate that this is a significant difference compared with the experimental model described in their study. Whether microglial activation would be comparable when induced by clearance/accumulation of extracellular alpha-synuclein or after constitutive alpha-synuclein overexpression remains unclear and could have probably been further discussed. Nonetheless, the (brief) discussion of caveats concerning the relevance of this model to pathogenetic processes in PD is now acceptable.

We thank the reviewer for confirming that this part of our discussion highlights key issues of our study and that this description is acceptable in his/her opinion.

Reviewer #3 (Remarks to the Author):

The authors have now addressed the questions raised by the reviewers. The data presented in the manuscript provide valuable findings that would be of interest to researchers studying the role of microglia in Parkinson disease, but also in other neurodegenerative diseases.

Minor points:

- Relating the protein, IBA1 still appears in lower case in several part of different figure (mostly in y-axis of graph). It should be capitalized. Ex: Fig 2g, 7a; Fig-supp 2,3,6,8
- At reverse, some genes in heatmaps are capitalized when they should be in lower case Ex: Fig 5b (CD206, TGF, IL1, IL6, CCL2), 6a (CD22)

We are thankful to the reviewer for the full appreciation of our work and its significance for neurodegenerative diseases. We have amended in the revised version the typos highlighted by the reviewer.